# Causal relevance of different blood pressure traits on risk of cardiovascular diseases: GWAS and Mendelian randomisation in 100,000 Chinese adults

Alfred Pozarickij[1,16], Wei Gan [1,2,16], Kuang Lin[1,16], Robert Clarke [1], Zammy Fairhurst-Hunter[1], Masaru Koido [3], Masahiro Kanai [4,5], Yukinori Okada [6,7,8,9], Yoichiro Kamatani [3], Derrick Bennett [1], Huaidong Du [1], Yiping Chen [1], Ling Yang [1], Daniel Avery [1], Yu Guo[10], Min Yu[11], Canqing Yu [12,13,14], Dan Schmidt Valle[1], Jun Lv[12,13,14], Junshi Chen[15], Richard Peto[1], Rory Collins[1], Liming Li [12,13,14,17] ✉, Zhengming Chen [1,17], Iona Y. Millwood [1,17], Robin G. Walters [1,17] ✉ & China Kadoorie Biobank Collaborative Group*

Elevated blood pressure (BP) is major risk factor for cardiovascular diseases (CVD). Genome-wide association studies (GWAS) conducted predominantly in populations of European ancestry have identified >2,000 BP-associated loci, but other ancestries have been less well-studied. We conducted GWAS of systolic, diastolic, pulse, and mean arterial BP in 100,453 Chinese adults. We identified 128 non-overlapping loci associated with one or more BP traits, including 74 newly-reported associations. Despite strong genetic correlations between populations, we identified appreciably higher heritability and larger variant effect sizes in Chinese compared with European or Japanese ancestry populations. Using instruments derived from these GWAS, multivariable Mendelian randomisation demonstrated that BP traits contribute differently to the causal associations of BP with CVD. In particular, only pulse pressure was independently causally associated with carotid plaque. These findings reinforce the need for studies in diverse populations to understand the genetic determinants of BP traits and their roles in disease risk.

Elevated blood pressure (BP) is a major modifiable cause of cardiovascular disease (CVD), including ischaemic heart disease (IHD) and stroke[1–3] In China, >4 million people develop CVD each year[4] and the disease burden has increased steadily in recent decades, partially reflecting the rising prevalence of hypertension and suboptimal treatment of individuals with high BP[5,6]. Randomised trials of several BP-lowering medications have demonstrated their efficacy for primary and secondary prevention of CVD[7–9]. Although widely used, the effectiveness of BP-lowering medications varies for different CVD types and according to overall proportions in the population of stroke versus IHD[10,11].

The most widely studied measures of BP are systolic BP (SBP) and diastolic BP (DBP), which reflect the force that the heart exerts on the arterial wall during the contraction and relaxation phases of the

A full list of affiliations appears at the end of the paper. *A list of authors and their affiliations appears at the end of the paper ✉e-mail: lmleeph@vip.163.com; robin.walters@ndph.ox.ac.uk

cardiac cycle, respectively. However, other measures of BP, including pulse pressure (PP) which reflects vascular stiffness, and mean arterial pressure (MAP), which reflects the average levels of blood pressure in arteries throughout the cardiac cycle, can provide additional insights into the determinants and consequences of BP.

BP is determined by a complex interaction between environmental and genetic factors. Observational studies and randomised trials have demonstrated that lifestyle (e.g. alcohol[12], smoking[13], diet[14–19]), and environmental (e.g. ambient temperature[20]) factors influence levels of BP. Large genome-wide association studies (GWAS) have identified >2,000 genetic variants associated with different BP traits, reflecting the high heritability of BP[21–27]. However, most of the previous GWAS have been conducted mainly among individuals of European ancestry, with more limited data from Chinese and other East Asian ancestry populations, in which the genetic architecture, environmental exposures, and healthcare provision for the treatment of high BP may differ substantially from those in Western populations. Moreover, most of the previous genetic studies have focused on SBP and DBP and, hence, relatively little is known about the genetic determinants of other BP traits including PP and MAP and their causal relevance for CVD.

In the present study, we performed GWAS of four different measures of BP (SBP, DBP, MAP and PP) in ~100 K adults from the China Kadoorie Biobank (CKB). Using summary statistics from European and Japanese ancestry cohorts, we compared BP heritability, genetic correlations, and SNP effect sizes on BP traits between Chinese, Japanese and European ancestry adults. In addition, using multivariable Mendelian randomisation (MR) approaches, we examined the contribution of different BP traits to the established causal relationships between BP and risk of different CVD types in Chinese and Japanese adults. We show that the causal relevance of BP traits varies for different CVD outcomes, informing future investigation of the underlying biological mechanisms and development of prevention and treatment strategies for CVD.

## Results
### Population characteristics
Among 100,453 CKB participants included in the GWAS (see Supplementary Fig. 1 for inclusion/exclusion criteria), the mean age (SD) was 53.7 (11) years, mean BMI was 23.7 (3.5) kg/m$^2$, 57.2% were women and 4.2% reported a prior history of CVD (Supplementary Data 1). Overall, men had a higher prevalence of hypertension (i.e. SBP ≥ 140 mmHg or DBP ≥ 90 mmHg at baseline recruitment, or self-reported prior diagnosis by a physician) than women (41.5% vs 36.6%). Among those with self-reported hypertension, approximately one-third reported the use of BP-lowering medication. For the studied traits (SBP, DBP, PP [defined as SBP − DBP] and MAP [defined as (2 x DBP + SBP)/3]), mean BP measures in the CKB population were 0.5–7.7 mmHg lower than in middle-aged populations of European ancestry (ICBP)[22], but were comparable to those in Japanese adults from Biobank Japan (BBJ)[21] (Supplementary Data 2). There were generally strong phenotypic correlations between BP phenotypes in CKB ($r^2 = 0.32–0.95$) (Supplementary Data 3), even for pairs of traits that might be thought of as orthogonal (DBP and PP, PP and MAP).

### Genome-wide association analyses
We conducted GWAS of four BP traits (SBP, DBP, PP, MAP) both with and without adjustment for BMI (i.e. 8 separate GWAS). Although our primary analyses included adjustment for BMI, we also conducted GWAS without adjustment for BMI to facilitate comparisons with previous GWAS of these traits which have, variously, included or omitted adjustment for BMI. We identified a total of 128 non-overlapping genomic regions representing 510 separate trait-locus associations at $P < 5 \times 10^{-8}$ (Table 1; Supplementary Data 4–12; Supplementary Figs. 2–5). Approximate conditional analyses identified 18 additional independent associations within 6 loci (Supplementary Data 13). Although the BMI-adjusted and unadjusted analyses differed somewhat in the loci identified (120 and 115 loci, respectively, with 90 identified in both analyses), these differences nearly all reflected small changes in statistical significance around the genome-wide significance threshold of $5 \times 10^{-8}$. There were only minor differences between the two models in the estimated effect sizes for all lead variants, with the exception of a signal at the *FTO* locus, which had a lower effect size for SBP, PP, and MAP in BMI-adjusted analysis (Supplementary Fig. 6).

Across the eight GWAS, there were 37 non-overlapping genomic regions corresponding to a total of 74 trait-specific associations which had not previously been reported in previous studies of BP in Europeans (ICBP[22,27]) or Japanese (BBJ[21,23,28–31]), or in the HGRI-EBI GWAS catalog[32] (see Supplementary Data 14). There were more newly-reported associations for the less well-studied trait MAP (22 and 20 with and without BMI adjustment, respectively) than for SBP (2 and 5), DBP (3 and 4), and PP (3 and 6) (Supplementary Data 4). One-third of these trait-locus associations were at loci previously reported as

## Table 1 | Genome-wide significant associations and heritabilities for blood pressure traits in CKB

| Trait | BMI adjustment | Loci | Newly reported | Heritability ($h^2_g$) | | |
|---|---|---|---|---|---|---|
| | | | | CKB | BBJ | ICBP |
| SBP | – | 66 | 5 | 0.162 (0.010) | 0.076 (0.007) | – |
| | + | 70 | 2 | 0.158 (0.012) | – | 0.129 (0.005) |
| | | **80** | **5** | | | |
| DBP | – | 65 | 9 | 0.164 (0.010) | 0.060 (0.006) | – |
| | + | 73 | 7 | 0.162 (0.011) | – | 0.129 (0.005) |
| | | **82** | **10** | | | |
| PP | – | 45 | 6 | 0.129 (0.009) | 0.051 (0.005) | – |
| | + | 45 | 3 | 0.124 (0.009) | – | 0.111 (0.004) |
| | | **51** | **7** | | | |
| MAP | – | 69 | 20 | 0.171 (0.010) | 0.072 (0.007) | – |
| | + | 77 | 22 | 0.168 (0.012) | – | 0.106 (0.009) |
| | | **86** | **27** | | | |
| Total | | **128** | **37** | | | |

Analyses were adjusted for sex, age, age$^2$, recruitment region, mean monthly external temperature, and BMI (as indicated). SNP-based heritability ($h^2_g$) was estimated using LD score regression. Numbers in brackets are standard errors. Numbers in bold represent non-overlapping loci.

*SBP* systolic blood pressure, *DBP* diastolic blood pressure, *PP* pulse pressure, *MAP* mean arterial pressure, *CKB* China Kadoorie Biobank, *BBJ* Biobank Japan, *ICBP* International Consortium of Blood Pressure.

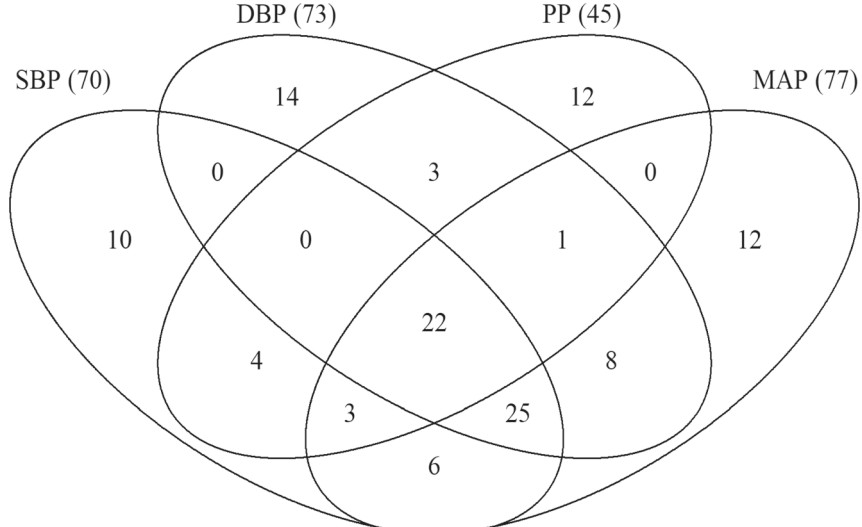

**Fig. 1 | Overlap of associations across BMI-adjusted blood pressure traits. a** Venn diagram of associated loci across BMI-adjusted blood pressure traits. The numbers in brackets indicate the total number of loci associated with that BP phenotype. **b** Genetic correlation between the 4 traits as assessed by LD score regression.

associated with at least one of the other BP phenotypes; after exclusion of these, 13 non-overlapping genomic regions were identified that had not previously been associated with any BP trait, comprising associations with one or more of SBP (2 and 4 with and without BMI adjustment, respectively), DBP (3 and 4), PP (1 and 3) and MAP (2 and 6). There were 64 loci which had not previously been associated with any BP phenotype in East Asian populations (Supplementary Data 4).

### Comparisons between BP traits
Many of the identified loci were associated with multiple traits (Fig. 1a; Supplementary Fig. 7). Out of 120 BP-associated loci from the BMI-adjusted analyses, 22 (18%) were associated with all four BP traits, 29 (24%) with three BP traits, 21 (17%) with two BP traits. Despite the simple arithmetical relationship between the traits, which was reflected in the corresponding variant effect sizes for those traits, 48 (40%) of the identified loci were associated with only one BP trait. The extent of overlap across traits was similar for the 115 associations from the BMI-unadjusted analyses. By contrast, of the 29 loci newly associated with BMI-adjusted traits, most of which were identified as associated with MAP, 24 (83%) were associated with one BP trait with the remaining 5 being associated with two traits (Supplementary Fig. 8).

The observational correlations between BP traits (Supplementary Data 3) were reflected in the between-trait genetic correlations (Fig. 1b). SBP and DBP showed strong genetic correlation with each other ($r_g$ = 0.849 [se 0.016]) and with MAP (which is derived from them), with SBP also strongly correlated with PP. By comparison, reflecting its derivation as the difference between SBP and DBP, PP had weaker genetic correlation with DBP ($r_g$ = 0.471 [se 0.049]; BMI-unadjusted: $r_g$ = 0.513 [0.039]) and with MAP ($r_g$ = 0.682 [0.033]; BMI-unadjusted: $r_g$ = 0.708 [0.026]). This was also reflected in comparisons

of effect sizes (Supplementary Fig. 9): variants associated with SBP, DBP, and MAP had effect sizes which were consistently correlated between traits, whereas variants identified in PP analyses showed very little correlation with the other traits. BP-associated variants consistently displayed larger effects on SBP than on the other traits, with Deming regression slopes of 1.76 [95%CI 1.72–1.80], 1.81 [1.73–1.90], and 1.44 [1.42–1.45], for comparisons with DBP, PP, and MAP, respectively. Similarly, variant effect sizes for MAP were larger than for DBP and PP (1.25 [1.24–1.26] and 1.22 [1.12–1.29], respectively).

### Comparisons between populations
We sought to replicate the 74 newly reported variant-trait associations within 37 loci by performing association analyses in BBJ, and by review of published ICBP summary statistics[22] (Supplementary Data 14). A total of 70 associations were directionally consistent in BBJ, with 30 (41%) being replicated at 5% false discovery rate (FDR); similarly, out of 18 CKB lead variants for which ICBP summary statistics were available, 11 associations were directionally concordant and 4 replicated at 5% FDR.

The replication analyses in BBJ consistently identified variant effect sizes of approximately one-third of those in CKB (Supplementary Fig. 10). To investigate further, while avoiding the potential influence of winners' curse[33], we compared effect sizes in CKB with those published for BBJ, for lead variants identified for each trait in ICBP analyses (Fig. 2). Effect sizes in CKB were again larger (1.7- to 2.1-fold) than for the corresponding associations in BBJ. The corresponding analyses comparing effect sizes in CKB and ICBP, for lead variants identified in BBJ, similarly showed 1.6- to 2.9-fold larger effect sizes in CKB. Consistent effect size differences were also found when separately comparing effect sizes in CKB participants recruited in

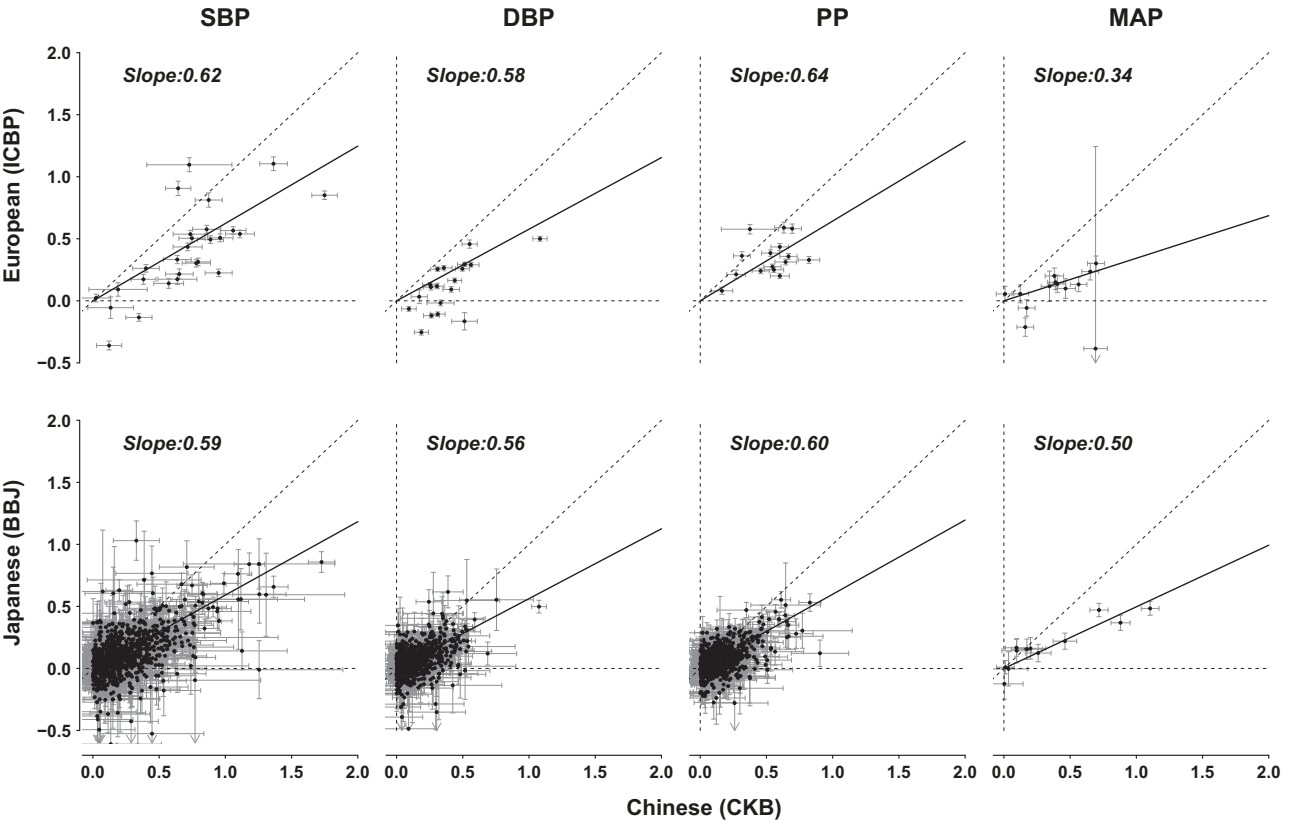

**Fig. 2 | Comparisons of variant effect sizes of blood pressure traits between CKB and ICBP (BMI-adjusted) and between CKB and BBJ (BMI-unadjusted).** Comparisons of CKB with each of ICBP and BBJ used genome-wide significant variants identified in BBJ and ICBP, respectively. Variant per-allele effects are in mmHg. Dashed diagonal lines are the identity line ($y = x$). Solid lines are the Deming regression line forced through the origin. Source data are provided as a Source Data file.

urban (1.3- to 3.0-fold larger effect sizes in CKB than in BBJ) or rural (1.6- to 2.6-fold larger effect sizes in CKB) regions (Supplementary Fig. 11).

Using summary statistics from each of CKB, BBJ and ICBP, estimates of narrow-sense heritability ($h^2$) using LD score regression were substantially and consistently higher in CKB compared with those observed for BBJ and ICBP, for all BP traits (Table 1). In particular, $h^2$ for all BP traits was over 2-fold greater in CKB than in BBJ. There were also modest but consistent $h^2$ differences between rural and urban CKB regions, with higher heritability in urban than rural regions: for each BP phenotype, $h^2$ in urban regions was 1.1- to 1.4-fold greater than in rural regions (Supplementary Fig. 12, Supplementary Data 15). Additional stratified heritability estimates by sex, age, and treatment group (Supplementary Data 16) showed no differences by sex or age, and adjustment for BMI had very little impact on BP trait heritability in any of these analyses. However, there was a substantial apparent difference in the heritability of SBP according to BP-lowering medication use (0.14 in those using medication vs 0.23 in others); a lesser, non-significant difference was observed for MAP (0.18 vs 0.25). We further explored the impact of BP-lowering medication by comparing effect sizes in CKB participants according to their self-reported medication use, for BP-associated variants previously reported in BBJ (Supplementary Fig. 13). For each trait other than PP, there was a 43% reduction in effect size amongst those using BP-lowering medication.

Despite these differences in variant effect size and heritability, for all BP traits (BMI-unadjusted) there was a high degree of between-population genetic correlation between CKB and BBJ (rg>0.870) as measured from summary effect sizes using both LD score regression and Popcorn[34] (Supplementary Data 17, Supplementary Data 18). Similarly, there was strong cross-ancestry genetic correlation between CKB and ICBP for all traits (BMI-adjusted, $r_g > 0.790$) (Supplementary Data 18). By comparison, genetic correlations between BBJ (BMI-unadjusted) and ICBP (BMI-adjusted) were somewhat lower ($r_g < 0.75$) for all traits.

## Associations of genetically predicted blood pressure with major vascular diseases

Using independent lead variants, we constructed weighted genetic scores (GSs) for each (BMI-adjusted) BP trait for use in MR analyses of causal relationships between BP traits and major CVD outcomes. To avoid bias towards the observational association due to overfitting of effect size estimates, weights were derived by jack-knifing. These GSs explained 3.1%, 3.1%, 2.0%, and 3.5% of trait variance for SBP, DBP, PP, and MAP, respectively, with a tendency for somewhat larger $r^2$ in rural than in urban regions (Supplementary Data 19). All instruments had an F-statistic of >100 within each region, indicating a low risk of weak instrument bias.

For all four traits, genetically-predicted BP showed strong positive associations with ischaemic stroke (IS), intracerebral haemorrhage (ICH), major coronary events (MCE) and carotid plaque (CP) (Fig. 3). For IS, the odds ratio (OR) per 1-SD higher SBP (22.3 mmHg) was 1.84 [95% CI 1.61–2.12], with other BP traits showing similar per 1-SD associations. However, when effect sizes were instead expressed per 5 mmHg higher BP, SBP had the weakest (1.15, 1.11–1.18), and DBP the strongest (1.28, 1.21–1.35) association with IS. For ICH, the effects of genetically determined BP were stronger than for IS, with SBP showing the strongest association per 1-SD higher BP (3.03, 2.53–3.64) but once again the smallest per 5 mmHg effect, for which DBP again showed the strongest association with ICH (1.52, 1.42–1.64). For MCE, the risk estimates were more modest, ranging from 1.35 to 1.37 per 1-SD higher

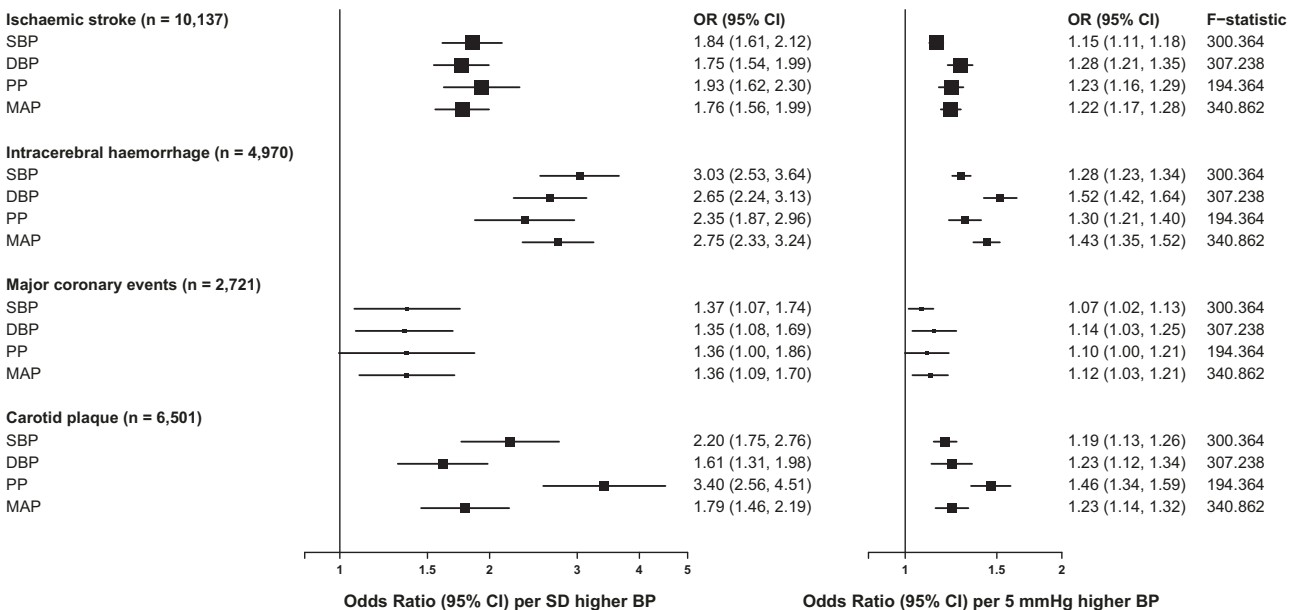

**Fig. 3 | Mendelian randomisation of blood pressure traits with risk of major cardiovascular diseases and subclinical atherosclerosis.** Effects are shown as odds ratios (95% CI) for disease risk per 1 SD higher blood pressure (left panel) or per 5 mmHg higher blood pressure (right panel). F-statistic was calculated as the mean across CKB regions. SBP, systolic blood pressure; DBP, diastolic blood pressure; PP, pulse pressure; MAP, mean arterial pressure. Source data are provided as a Source Data file.

BP. While all four BP traits were significantly associated with CP, genetically predicted PP showed the strongest association per 1-SD (3.40, 2.56–4.51) or 5 mmHg (1.46, 1.34–1.59).

To assess whether these associations might be subject to bias due to pleiotropy, we also conducted two-sample inverse variance weighted MR (IVW-MR) and MR-Egger[35], using summary statistics from both CKB[36] and BBJ[28] (Supplementary Data 20). Associations of each BP trait with each outcome were consistent with those from the GS-based analyses, with no evidence for unbalanced horizontal pleiotropy.

**Multi-variable Mendelian randomisation**

To explore the extent to which associations of specific BP traits are due to direct causal associations with disease, rather than arising out of a strong genetic correlation and/or collinearity with one or more other traits, we performed MVMR, in which pairs of instruments for different BP traits were included in a single model (Supplementary Fig. 14). Each of these analyses, using slightly modified GSs for each trait, gave single-variable MR estimates which were consistent with the preceding trait-specific MR analyses (Supplementary Fig. 15, Fig. 3); however, when traits were were mutually adjusted for each other, changes in effect estimates were observed which are interpreted as reflecting the causal relevance of individual BP traits independent of the other BP trait included in the analysis (Fig. 4, Supplementary Fig. 15, Supplementary Data 21).

The GSs for these analyses comprised variants at all loci that were associated with either of the pair of traits under investigation and, therefore, included a larger number of variants and explained a higher proportion of trait variance (2.4–3.8%) than the GSs constructed for single-variable MR (Supplementary Data 21). To ensure MVMR results were not biased by selection of variants more strongly associated with one of a pair of traits, we performed sensitivity analyses using two further sets of GSs for which variants at a locus were selected according to the strength of their association with one or other of the traits being analysed; these results were very similar to those from the main analyses (Supplementary Figs. 16–17).

For most pairs of traits, the conditional F-statistic was greater than 10, indicating a low risk of weak instrument bias; however, when MAP was analysed with either SBP or DBP, conditional F-statistics were lower than 10 (Supplementary Data 21), so that caution is required when interpreting the results of analyses using these pairings. This MAP-SBP and MAP-DBP collinearity was reflected in wide confidence intervals in MVMR analyses (Supplementary Figs. 15–18). We attempted MVMR including all four traits in a single model, but these analyses gave very low conditional F-statistics for each trait and invariably yielded unstable estimates.

For IS, mutual adjustment of SBP and DBP resulted in partial attenuation of their effects observed in single-variable MR, giving ORs of 1.40 [0.97–2.01] and 1.34 [0.96–1.87] per 1-SD higher BP, respectively, while neither PP nor MAP contributed independently to IS risk after adjusting for SBP (Fig. 4a). A similar overall pattern was observed for MCE, although there was an insufficient number of cases to reliably distinguish which BP traits were independent risk factors (Fig. 4b). For ICH, SBP remained independently causally associated with increased risk in multivariable MR, with each of DBP, PP, and MAP being attenuated to the null after adjustment for SBP (Fig. 4c); in these models, adjusted ORs per 1-SD SBP were between 2.29 [1.42–3.68] and 3.90 [2.42–6.31], consistent with the estimate from single-variable MR.

MVMR gave results for CP which were clearly different from those for the other disease outcomes. Analyses of SBP with each of DBP or MAP gave unstable models in which the effect of SBP was increased while DBP and MAP appeared to have protective effects, while mutual adjustment of PP and SBP attenuated the association with SBP to the null, while that with PP was not appreciably unaffected (Fig. 4d). Further models in which PP was mutually adjusted for DBP or for MAP yielded consistent results in which effects of DBP and MAP were attenuated to the null, while PP retained strong association with risk of CP with ORs similar to those from single-variable MR (Supplementary Fig. 15).

To address potential bias due to sample overlap, we performed two-sample MVMR, either using variant effect sizes for BP (this study) together with previously published summary statistics in BBJ for CVD outcomes[28] or, for CP, by splitting the CKB dataset into 2 non-overlapping sub-samples (Supplementary Fig. 18, Supplementary Data 22). In these analyses, several pairs of traits (SBP-DBP, SBP-MAP, DBP-MAP) yielded conditional F-statistics that were substantially lower than 10 so that the effect estimates were subject to weak instrument bias. Nevertheless, for those trait pairs with F-statistics >10, the effect

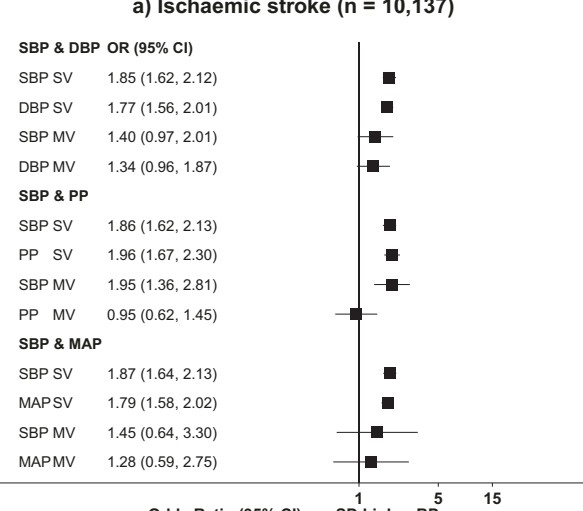

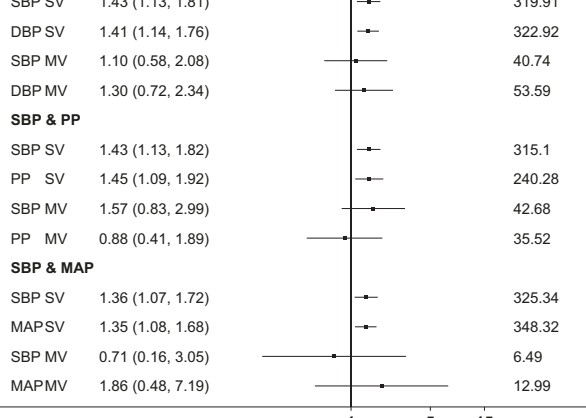

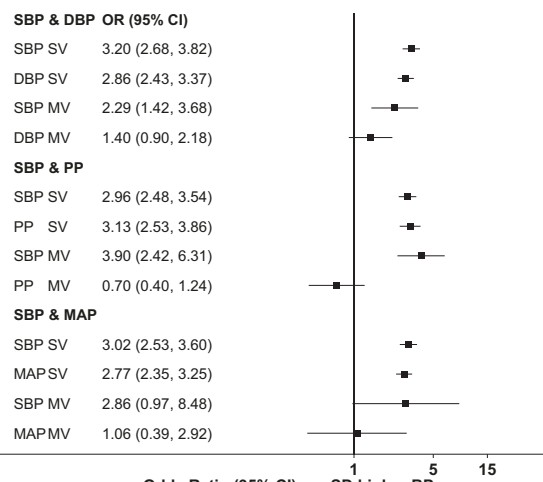

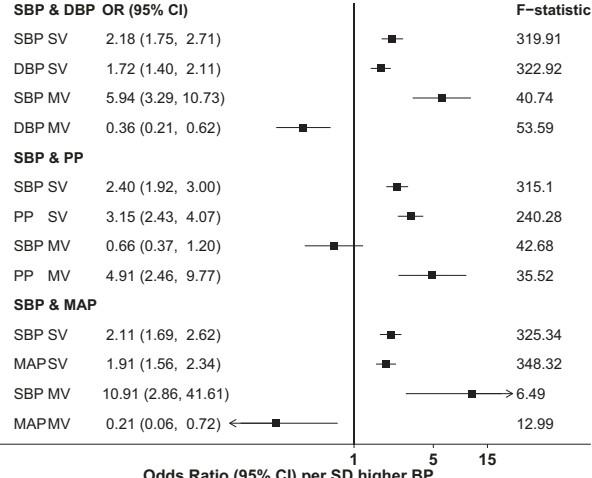

**Fig. 4 | Multivariable Mendelian randomisation of blood pressure traits with risk of major cardiovascular diseases and subclinical atherosclerosis.** Effects are shown as odds ratios (95% CI) per 1-SD higher blood pressure for (**a**) ischaemic stroke; (**b**) major coronary events; (**c**) intracerebral haemorrhage and (**d**) carotid plaque. Overlapping loci from two traits were merged and, in cases where there was more than one lead variant at a locus, the association with the lowest *P*-value was used to determine which variant to include when constructing the genetic scores. SBP systolic blood pressure, DBP diastolic blood pressure, PP pulse pressure, MAP mean arterial pressure, SV single variable MR, MV multi-variable MR. F-statistic was calculated as the mean across CKB regions. F-statistic for MV indicates the conditional F-statistic. Source data are provided as a Source Data file.

estimates were largely consistent with those from the within-CKB analyses, with the exception that two-sample MVMR analyses suggested that both SBP and DBP make a causal contribution to ICH risk (Supplementary Fig. 18c, Supplementary Data 22).

## Discussion

In this large study of the genomics of four BP measures in Chinese adults, we identified a total of 128 genetic loci significantly associated with BP, of which 64 had not previously been associated with BP in East Asians, and 13 were not previously associated with BP in any ancestry. We identified 74 previously unreported trait-locus associations, of which a large proportion were for MAP, likely reflecting the limited previous studies investigating MAP. About half of the newly identified associations were replicated in analyses of Japanese and/or European adults. For all BP traits, narrow sense heritability and variant effect sizes were greater in Chinese compared with other populations, especially Japanese. GSs derived from these loci explained 2.0% to 3.5% of the variance of the different BP traits. In MR analyses, all four BP

traits showed highly significant associations with risks of different CVD types, with comparable per 1-SD risk estimates across traits, which were greater for ICH than for IS and MCE. For CP, however, PP was more strongly associated than other BP traits, and was independently associated with carotid plaque after accounting for the other BP traits.

BMI is a strong risk factor for BP in both observational and MR studies[37-39]. Previous GWAS of BP have variously included or omitted adjustment for BMI, so to improve comparability with other studies we performed GWAS both with and without such adjustment. Adjustment for BMI had little impact on SNP effect sizes (with the exception of the *FTO* locus), and did not affect the significance of association for the majority of BP loci. Indeed, adjustment for BMI somewhat increased the number of genomic regions associated with different BP phenotypes. We infer that, although BMI overall has an impact on BP[37], individual adiposity-associated variants make only a very modest contribution to BP.

More than 2,000 genetic associations with various BP phenotypes have previously been identified[21-23,26-30,40]. The loci corresponding to

these associations overlap with signals for many other traits (e.g. obesity, lipids) and CVD outcomes[22], and can potentially inform the development of drugs for treatment of hypertension. Pathway analyses have identified enrichment of BP signals at loci with roles in the structure of arterial walls, and have indicated that TGF-β and Notch signalling pathways play an important role in BP regulation. Previous BP GWAS studies have been primarily conducted in individuals of European ancestry[22,25–27,40], raising questions about the relevance of these findings to other ancestry populations, but more recent analyses have improved our understanding of the genetic architecture of BP in other ancestries, and have established that many BP associations are replicated in diverse populations[21,23,28–30]. Nevertheless, we have demonstrated that large studies in non-European populations provide continuing opportunities for discovery of genetic associations with BP. In the present study, we newly identified 13 BP loci, despite having a sample size that was approximately one-tenth that of previous studies of European individuals[22,27,40] and similar to those of previous studies of East-Asian populations[21,23,28–30].

SBP and DBP are the most widely used measures of BP in clinical practice. Although both PP and MAP are directly derived from SBP and DBP, they reflect discrete aspects of biology. PP (calculated as the difference between SBP and DBP) potentially reflects differences in cardiac structure or wall thickness caused by sustained hypertension, valvular regurgitation, age-related aortic stiffness, or in rarer instances severe iron deficiency anaemia or hyperthyroidism[41,42]. MAP (less often included in genetic analyses) represents the mean pressure during a cardiac cycle and is influenced by cardiac output, systemic vascular resistance, volume and viscosity of circulating blood, and the elasticity of vessel walls[43]. Consistent with previous reports in different ancestries[44,45], and reflecting their arithmetical relationships, we found strong genetic correlation between these 4 different BP traits. Accordingly, the majority (~60%) of BP loci in the present analyses were associated with multiple BP phenotypes. Nevertheless, 24 of the 128 loci would not have been identified by considering only SBP and DBP, highlighting the value of conducting analyses of multiple related traits.

Previous GWAS of BP have highlighted the highly heritable nature of BP, with estimates of narrow sense heritability (due to additive genetic effects) for SBP and DBP in the range of 17–52%[46]. We observed systematically lower heritability for all traits in European and Japanese populations compared to Chinese. In particular, heritability estimates in BBJ were less than half those in CKB. By contrast, LD score estimates of $h^2$ for SBP and DBP reported for the Taiwan Biobank (TWB)[29] were similar to those for CKB. Although our estimates of heritability based on CKB, BBJ, and ICBP summary statistics using LD score regression were lower than in previous reports (Table 1), this is likely to be because these reflect common variation only; substantially higher $h^2$ estimates (19–21%) were reported by ICBP for UK Biobank using REML as implemented in BOLT-LMM[47], and we found comparable increases in heritability estimates in CKB (20–27%) when using REML (Supplementary Data 23).

In addition to lower heritability estimates compared to CKB, variant effect size estimates were also substantially lower in both ICBP and BBJ (Fig. 2, Supplementary Fig. 11). Although several previous studies have also examined differences in heritability across populations[21,26], we are unaware of studies that have reported systematic differences in variant effect size on the scale observed. Nevertheless, despite these large differences in estimated heritability and effect sizes, we found strong cross-population genetic correlation of BP traits across ancestries and between East Asian populations, consistent with previous findings for several other diseases and complex traits[34,48,49]. This suggests that the observed differences in heritability do not reflect major differences in the genetic architecture of BP traits. Rather, we suggest that external factors may modify the extent to which genetic variation influences BP, leading to changes in variant effects and, consequently, heritability.

There are multiple differences between the CKB, BBJ and ICBP populations that might impact on effect size and heritability. Specific environmental factors such as differences in lifestyle, including dietary patterns, may contribute to differences in measured heritability and variant effect sizes. Sodium intake is associated with hypertension[16,50] in addition to higher risks of chronic diseases[17–19,51] and salt intake in China has been estimated to be among the highest in the world[51]. Alternatively, lower environmental variability has the potential to increase measured heritability, by increasing the proportion of phenotypic variance accounted for by genetics. Stratified heritability estimates in CKB found no appreciable effects of region, sex or age, but did reveal a substantial difference in heritability of SBP (and also MAP, to a a lesser extent) according to BP-lowering medication use. Similarly, medication use corresponded to a reduction in variant effect size for each of SBP, DBP, MAP (but not PP).

Taken together, these effects of medication use on heritability and effect size provide a potential explanation for the observed differences between populations. CKB is a population-based cohort (as is TWB), while BBJ is based on a combination of different hospital-based disease cohorts, who may have different BP characteristics from healthy adults. Of note, only 12% of CKB participants in this study (and 14% of TWB participants[31]) were taking BP-lowering medication at the time of BP measurement, compared with 34% of BBJ. By contrast with Japanese and European-ancestry individuals, hypertension is still poorly detected, treated and controlled in China[5]. A nationally representative sample of ~450 K individuals in China reported that around half of hypertensive patients did not take prescribed BP-lowering medications, and BP was controlled for only one in six of those treated[6]; similarly, in CKB BP was successfully controlled only in approximately 5% of participants with hypertension[2,52]. The substantial reduction in effect size of BP variants in CKB participants taking anti-hypertensive medication suggests that differences in medication use may at least partly account for the observed differences between CKB and BBJ. Thus, although all analyses in ICBP, BBJ, and CKB included adjustment for use of BP-lowering medication, the findings of the present study may reflect in part the impact of performing analyses in largely untreated populations.

Although observational associations of higher levels of BP with increased CVD risk are well established, the strong correlation between individual BP traits makes it very difficult to establish the independent causal relevance of individual BP traits. Similarly, previous MR studies have demonstrated apparent causal effects of multiple BP traits on different CVD types[53–55]. Consistent with these, using strong genetic instruments derived from genome-wide significant associations, all four BP traits showed similar per-SD associations with risk of different CVD types in Chinese, with SBP being more strongly associated with increased risk than DBP. Notably, however, when expressed per unit change in SBP (i.e. per 5 mmHg), SBP showed the weakest associations with CVD outcomes, while DBP had the strongest association. The odds ratios from these analyses were consistent with those from observational analyses in CKB[2]. By contrast, for CP the strongest association was consistently observed for PP, suggesting a potentially different causal contribution to CP risk than for other CVD outcomes.

Although suggestive, such MR analyses do not provide conclusive evidence about which traits have a true causal relationship with different CVD-related outcomes. The strong correlations between variant effect sizes for different BP measures (Supplementary Fig. 9) suggest that a GS for one trait is also an effective instrument for another and cannot completely discriminate between them. For this reason we employed MVMR, in which the association with one instrument is estimated conditional on a second instrument for a different trait, but with both instruments using identical genetic variants. We observed mutual partial attenuation by SBP and DBP for risk of IS (and also for MCE, although this analysis was less well-powered), suggesting that SBP and DBP (and/or their related biological processes) each independently contribute to CVD risk.

Very few previous studies have assessed the independent effects of different BP traits on ICH risk, and none have had sufficient power to investigate the independent effects of BP traits. In MVMR analyses using individual-level data, the associations of DBP, MAP, and PP instruments with ICH were completely attenuated to the null by the inclusion of an SBP instrument, whose association was largely unaffected in the joint analysis; however, two-sample approaches were less clear, with mutual partial attenuation by SBP and DBP for ICH, as for IS. However, interpretation of the two-sample MVMR is complicated by the weak instruments for several trait pairs. Furthermore, the ICH cases included in the BBJ GWAS were recruited subsequent to the ICH event, and are likely to have a different spectrum of case severity and subtype, possibly accounting for differences compared with the CKB-only analyses. Nevertheless, in all MVMR analyses it was clear that PP made no independent contribution to ICH risk.

By contrast, we identified a specific causal association of PP (and/ or related biological processes) with CP, which was unaffected by adjustment for instruments for other BP traits, whose associations were attenuated to the null by the inclusion of a PP instrument. This is consistent with previous observational studies which reported strong associations of PP with CP, which may relate to arterial remodelling in response to turbulence at the bifurcation of the common carotid arteries, initiating the development of carotid plaques[56,57].

Previous MVMR of BP traits have been limited to two-sample studies of SBP and DBP, based on summary statistics for ischaemic CVD in populations of European ancestry[53,58], in whom the proportions of different subtypes of CVD differ substantially from those in CKB (e.g. much higher frequency of cardioembolic stroke)[59]. Both previous studies found that effects due to DBP were attenuated more strongly than those from SBP, but both studies had wide confidence intervals and are consistent with our findings that both SBP and DBP contribute to IS risk. Importantly, our ICH and CP analyses demonstrate that we were able to reliably identify (or exclude) specific independent effects of individual BP traits on disease risk. The absence of such a finding in the better-powered IS analysis (with approximately twice as many cases as ICH or CP), suggests that the mutual attenuation by SBP and DBP instruments for IS reflects a genuine causal contribution by both traits, or by one or more underlying biological processes that affect both of them.

The present study has several strengths, including prospective study design, standardised data collection across the entire study population, comprehensive capture of disease outcomes with validation of diagnoses through retrieval of medical notes including details of imaging and other investigations, large sample size in an understudied population with a range of different CVD types, and availability of individual participant data. We were able to perform well-powered GWAS with replication in independent populations and, in addition to comparisons of GWAS of multiple BP traits, we performed systematic assessment of variant effect sizes and heritability across different ancestries. This enabled us to construct strong genetic instruments that could be used for MR and MVMR analyses of different BP traits with several CVD outcomes, with no evidence of horizontal pleiotropy. Nevertheless, the study had several limitations: GWAS discovery was conducted within the study population used for the main MR analyses, although restricting instruments to genome-wide significant loci limited the potential for weak instrument bias; there remained appreciable collinearity between genetic instruments, leading to unreliable MVMR estimates in some analyses due to weak instruments, but these could be identified from their small conditional F-statistics; and we were unable to attempt replication of our CP results due to there being no available well-powered East Asian GWAS summary statistics for use in two-sample MVMR. It would be of value to explore the mechanisms underlying the newly identified BP associations by integrating results with other omics resources. However, at present we are limited by the lack of suitable EAS datasets for these various analyses, e.g. GTEx only has 12 donors of Asian decent[60].

Overall, this large genetic study of Chinese adults identified a total of 128 genetic loci associated with BP traits, of which 13 had not been previously reported for any BP trait. Across three populations, there were substantial differences in heritability and variant effects for all BP traits. Although all four BP traits showed highly significant associations with risks of different CVD types in MR analyses, their causal relevance differed, with SBP and DBP making independent contributions to IS risk, SBP and possibly DBP being causal for ICH, and PP being independently associated with CP. The findings of this study enhance our understanding of the genetic architecture of BP traits in different ancestries, and of the physiological processes responsible for BP-associated increases in CVD risk.

## Methods
### Study population
The China Kadoorie Biobank (CKB) is a prospective study of >513,000 Chinese individuals aged 30-79 years recruited from 10 geographically defined regions during 2004-08[61–63]. The baseline survey collected extensive information about participants' socio-demographic status, lifestyle factors, environmental exposures, medical history (including medication use) and physical characteristics, through an interviewer-administered questionnaire and physical measurements (Supplementary Data 1). Each participant also provided a blood sample for long-term storage. Data on disease outcomes were identified from claims to the national health insurance system and from local chronic disease registries and disease surveillance point system death registries. All events were ICD-10 coded by trained staff blinded to baseline information. Active annual follow-up through local residential and administrative records confirms the participant's vital status. After the baseline survey, 5% of randomly selected surviving participants were re-surveyed periodically. In the 2013-14 resurvey, as well as repeating measurements made at baseline, carotid ultrasound examination was performed to assess cIMT and CP. The presence of CP was defined as focal thickening or protrusion from the artery wall into the lumen with carotid intima-media thickness >1.5 mm[64]. All participants provided written informed consent at each survey visit. Ethical approval was obtained from the Oxford Tropical Research Ethics Committee, University of Oxford (UK, 025-04) and the Ethical Review Committees of the Chinese Centre for Disease Control and Prevention (Beijing, China, 005/2004), Chinese Academy of Medical Sciences, and the Institutional Review Board (IRB) at Peking University.

### Blood pressure measurement
At baseline, BP was measured twice with a 5-min interval between measurements using Omron UA-779 digital sphygmomanometers (A&D Instruments; Abingdon, UK) that were regularly maintained and calibrated. A third BP measurement was recorded only if the difference between the previous two SBP measurements was greater than 10 mmHg. The mean of the last two readings was recorded and used for analyses[2]. Participants who reported using BP-lowering medication at baseline had their recorded BP adjusted by adding a constant value for SBP (15 mmHg) and DBP (10 mmHg). Using these adjusted values, PP was calculated as (SBP – DBP) and MAP was calculated as ([2 × DBP] + SBP)/3.

BP phenotypes for GWAS were derived from the whole CKB cohort of 513,214 individuals, before exclusions (Supplementary Fig. 1). We performed all adjustments for covariates and data transformations in the full CKB cohort prior to genetic analyses, so that these adjustments were not distorted by the non-random nature of those selected for genotyping[36]. Two individuals with missing BMI measurements and 371 individuals with BP measurements greater than 5 SDs from the population mean were excluded. After linear regression of each BP trait on age, age², sex, CKB study region, mean monthly outdoor temperature (temperatures below 5 °C were set to 5 °C)[65], and BMI, an additional 321 individuals were excluded with residuals greater than 5 standard deviations from the mean for one or more measures of BP.

The regressions were repeated for the remaining 100,453 participants, with/without adjustment for BMI, and residuals from the regression for these participants were used in association analyses.

## Genotyping and SNP quality control

Genotyping and quality control have been described in detail elsewhere[36]. After exclusions of samples with low call rate (<95%), high heterozygosity (3 SD above the mean), sex mismatch, ancestry outliers, XY aneuploidy, potential linkage errors, and missing/withdrawn consent, genotyping data were available for 100,706 participants. After imputation into the 1000 Genomes Phase 3 reference, variants with MAF < 0.005 and imputation accuracy ($r^2$INFO) < 0.3 were excluded. For analyses within each CKB geographic region separately, an additional 6,096 individuals with non-local ancestry were excluded[36]. All genomic locations are specified using the GRCh37 build.

## CVD outcomes

Causal relationships were assessed between BP phenotypes and incident CVD events, in which cases were defined according to the first CVD event (multiple first events of different types on the same day were excluded): 10,137 ischaemic stroke cases (IS, ICD-10: I63); 4,970 intracerebral haemorrhage cases (ICH, ICD-10: I61); and 2,721 major coronary events (MCE) comprising myocardial infarction (MI, ICD-10: I21-I23) or fatal ischaemic heart disease (IHD, ICD-10: I20- I25). 72,587 common controls with no CVD events were drawn from a population representative subset of genotyped participants[36]. For analyses of carotid plaque (CP, 6,501 cases), controls were all other genotyped second resurvey participants with available carotid ultrasound data.

## GWAS

Genome-wide association analyses using residualised BP phenotypes for 100,453 genotyped participants were conducted using BOLT-LMM version 2.3.2[47], with array version as covariate. As a sensitivity analysis, GWAS were performed in each CKB region separately and summary statistics were meta-analysed using an inverse-variance-weighted fixed-effect model in METAL[66].

We estimated the genomic inflation (λGC) of CKB BP GWAS to range from 1.20 to 1.31 (mean λGC = 1.261). Using pre-computed LD scores for East Asian and European ancestries (https://data.broadinstitute.org/alkesgroup/LDSCORE/), LD score regression intercepts (standard errors) for BMI-adjusted models were 1.056 (0.010), 1.054 (0.011), 1.039 (0.009), and 1.060 (0.011) for SBP, DBP, PP, and MAP, respectively. These were substantially smaller than the mean $χ^2$ and were comparable with those for other studies (Supplementary Data 24), indicating no substantial inflation except for that due to the polygenic nature of BP traits. Assessment of genomic inflation for BMI-unadjusted analyses was similar.

## Heritability and genetic correlation

Estimates of narrow-sense heritability and between-trait genetic correlation used LD score regression applied to summary statistics from this study, BBJ[21], and ICBP[22] for ~1 M HapMap3 SNPs with MAF > 0.05, INFO > 30 and $χ^2$ < 30 and excluding variants in the HLA region (chr6:21Mb-41Mb)[67]. LD score regression was also used for additional comparisons between CKB and BBJ. Cross-ancestry/cross-population genetic correlations between studies based on summary statistics for each trait were estimated using Popcorn[34] and cross-population LD scores for variants with MAF > 0.01 in the East Asian populations of the 1000 Genomes Project Phase 3[68]. Estimates of heritability in subsets stratified by recruitment region, sex, age, and BP-lowering medication used REML as implemented in BOLT-LMM[47].

## Locus identification

Genomic regions defined by genome-wide significant variants ($P < 5 \times 10^{-8}$) were defined by LD- based clumping in PLINK[69] (initial window ± 10Mbp, $P < 0.05$, LD $r^2 > 0.05$) using an internal LD reference of 10,000 unrelated CKB participants with imputed genotype probabilities converted to best-guess genotypes. We observed that, for three loci, the lead variant lay out outside the reported clumped range (all variants in LD lay either proximal or distal to the lead variant); in these instances we extended the clumped region to include the lead SNP. All locus boundaries were then extended by an additional 1Kbp in each direction. Overlapping loci were merged and the variant with the lowest P-value was identified as the sentinel. Conditional analyses to identify independent associations within a locus were performed using the stepwise model selection procedure in GCTA[70]. For comparisons of loci across traits, all overlapping loci identified for any trait were merged into single larger genomic regions which were then treated as single loci.

Locus novelty was assessed by testing whether any previous lead variant from previous studies of BP[21–23,25,28,29,36], or reported in the HGRI-EBI GWAS catalog[30] (accessed on 25/07/2023), lay within the locus. Assessment of novelty in East Asians used only lead variants from the relevant studies[21,23,28,29]. Replication of 74 newly identified associations was conducted using an independent sample from BioBank Japan (BBJ, $N_{max}$ = 133,567) or in silico using existing publically available ICBP summary statistics[22,25]. In the Japanese sample, phenotype residuals were obtained after adjusting for sex, age, age[2], first 10 principal components, disease status (affected versus non-affected) for each of the 47 target diseases in the BBJ, and with or without adjustment for BMI (Supplementary Fig. 19). Replication was assessed at Benjamini-Hochberg 5% false discovery rate.

## Comparison of variant effects across BP traits, ancestries, and subgroups

Comparisons of variant effects, between BMI-adjusted and BMI-unadjusted models, between traits, between population subgroups, and between 3 populations, were each assessed using Deming regression[71] as implemented in R. For comparisons between pairs of populations with the third excluded, the variants compared were lead variants as reported by the excluded study. BBJ GWAS summary statistics[23] for all phenotypes were downloaded from http://jenger.riken.jp/en/result. ICBP GWAS summary statistics were downloaded from http://biota.osc.ox.ac.uk) and https://ftp.ncbi.nlm.nih.gov/dbgap/studies/phs000585/analyses. BBJ summary statistics (expressed in SD units) were converted to mmHg using the phenotype-specific SD (Supplementary Data 2).

## Mendelian randomisation

MR was conducted using genetic scores (GS) constructed from the sentinel variants at each associated locus in the BMI-adjusted analyses. Variant effect sizes re-estimated using a sample of ~70 K unrelated individuals (king-cutoff 0.05) were used to construct GSs for the remaining set of 30 K individuals. For the ~70 K unrelated subset, we further re-estimated effect sizes by performing 100-fold jack-knifing, i.e. the effect sizes were estimated in 99% of the sample and used to construct GSs for the remaining 1%. For MR, the score for each trait was the sum of the dosages of risk alleles for that trait, weighted by the effect size from the corresponding GWAS. For MVMR with a pair of traits, a single sentinel variant was selected for each locus associated with either of the traits (overlapping loci were merged and treated as a single locus); where there was a choice of two sentinel variants (one for each trait), the variant with the lowest P-value was selected. The score for each trait was the sum of the dosage of risk alleles for the selected variants, weighted by the effect sizes for those variants from the corresponding GWAS. To assess potential bias due to greater power in the GWAS for one of the pair of traits, two alternative scores were derived in which selected variants were always those most strongly associated with one of the traits being analysed.

Linear regression of each BP trait on its GS, with BMI, sex, age, $age^2$, and regional principal components as covariates, was performed to derive the coefficient for the GS; this was used to scale individual GS values to derive genetically predicted BP traits. Associations of these genetically predicted exposures with binary vascular events were then assessed using logistic regression, with adjustment for sex, age, and $age^2$. For MVMR, pairs of genetically predicted BP traits were mutually adjusted to give causal estimates for each trait independent of the other. Both MR and MVMR were performed separately in each CKB recruitment region, and the estimates from each region were combined using inverse-variance weighted fixed-effect meta-analysis.

For each GS, the F-statistic and variance explained (estimated as the partial $r^2$ as the proportional reduction of the error sum of squares (SSE) of a full regression model including the GS compared to a reduced model without the GS) were as calculated by the R function anova(). For the calculation of conditional F-statistic and partial $r^2$ in MVMR, the models included the GS for the second trait in the analysis.

Two-sample MR and MVMR used summary statistics for IS, ICH, and MI from CKB[36] and BBJ[28] (downloaded from pheweb.jp). Variant effect estimates for CP in 21,534 CKB second resurvey participants were derived by logistic regression with age, $age^2$, sex, CKB region, and 11 national principal components as covariates. For two-sample MR and MVMR of CP we re-estimated BP variant effects in a sample that excluded those individuals used for CP. We used the inverse variance weighted method as implemented in the R package MendelianRandomization[72]. MR-Egger as implemented in the same R package was used to test for unbalanced horizontal pleiotropy.

### Reporting summary
Further information on research design is available in the Nature Portfolio Reporting Summary linked to this article.

## Data availability
The China Kadoorie Biobank (CKB) is a global resource for the investigation of lifestyle, environmental, blood biochemical and genetic factors as determinants of common diseases. The CKB study group is committed to making the cohort data available to the scientific community in China, the UK and worldwide to advance knowledge about the causes, prevention and treatment of disease. For full details of what data are currently available to open access users and how to apply for it, visit: https://www.ckbiobank.org/data-access.

Summary statistics for this study are available for download from GWAS Catalog (https://ftp.ebi.ac.uk/pub/databases/gwas/summary_statistics/GCST90335001-GCST90336000/) under accession codes GCST90335162, GCST90335163, GCST90335164, GCST90335165, GCST90335166, GCST90335167, GCST90335168, GCST90335169, and from the CKB PheWeb browser at pheweb.ckbiobank.org. Source data are provided with this paper.

Researchers who are interested in obtaining the raw data from the China Kadoorie Biobank study that underlines this paper should contact ckbaccess@ndph.ox.ac.uk. A research proposal will be requested to ensure that any analysis is performed by bona fide researchers and - where data is not currently available to open access researchers - is restricted to the topic covered in this paper. Source data are provided with this paper.

## Code availability
Code used for the data analyses in this study can be made available by contacting the corresponding authors. Access to code will be granted for requests for academic use within 4 weeks of application.

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

## Acknowledgements
The chief acknowledgment is to the participants, the project staff, and the China National Centre for Disease Control and Prevention (CDC) and its regional offices for assisting with the fieldwork. We thank Judith Mackay in Hong Kong; Yu Wang, Gonghuan Yang, Zhengfu Qiang, Lin Feng, Maigeng Zhou, Wenhua Zhao, Yan Zhang and Zheng Bian in China CDC; Lingzhi Kong, Xiucheng Yu, and Kun Li in the Chinese Ministry of Health; and Garry Lancaster, Sarah Clark, Martin Radley, Mike Hill, Hongchao Pan, and Jill Boreham in the CTSU, Oxford, for assisting with the design, planning, organisation, and conduct of the study. Members of the China Kadoorie Collaborative Group are listed in the supplementary material. The CKB baseline survey and the first re-survey were supported by the Kadoorie Charitable Foundation in Hong Kong. Long-term follow-up of the CKB study was supported by grants to Z.C. at Oxford University from the Wellcome Trust (212946/Z/18/Z, 202922/Z/16/Z, 104085/Z/14/Z, 088158/Z/09/Z) and grants to L.L. from the National Key Research and Development Program of China (2016YFC0900500, 2016YFC0900501, 2016YFC0900504, 2016YFC1303904) and the National Natural Science Foundation of China (91843302). DNA extraction and genotyping was supported by grants to Z.C. from GlaxoSmithKline and the UK Medical Research Council (MC-PC- 13049, MC-PC-14135). The project was supported by core funding from the UK Medical Research Council (MC_UU_00017/1,MC_UU_12026/2 MC_U137686851), Cancer Research UK (C16077/A29186; C500/A16896) and the British Heart Foundation (CH/1996001/9454) to the Clinical Trial Service Unit and Epidemiological Studies Unit at Oxford University. Computation used the Oxford Biomedical Research Computing (BMRC) facility, a joint development between the Wellcome Centre for Human Genetics and the Big Data Institute supported by Health Data Research UK and the NIHR Oxford Biomedical Research Centre; the views expressed are those of the author(s) and not necessarily those of the NHS, the NIHR, or the Department of Health.

## Author contributions
A.P., W.G., K.L., M.Ko., M.Ka., Y.O., Y.K. analysed the data. A.P. drafted the manuscript. A.P., W.G., K.L., R.C., Z.F.H., D.B., I.Y.M., L.L., Z.C., and R.G.W. contributed to the conception of this paper, interpretation of the results, and the revision of the manuscript. R.G.W., L.L., and Z.C. designed the study. K.L., M.Y., Y.G., J.L., C.Y., D.A., D.S.V., L.L., H.D., Y.C., L.Y., J.C., R.P., R.C., Z.C., I.Y.M., and R.G.W. contributed to data acquisition. All authors critically reviewed the manuscript and approved the final submission.

## Competing interests
The authors declare no competing interests.

## Additional information

¹Clinical Trial Service Unit and Epidemiological Studies Unit (CTSU), Nuffield Department of Population Health, University of Oxford, Oxford, UK. ²Human Genetics Centre of Excellence, Novo Nordisk Research Centre Oxford, Innovation Building, Old Road Campus, Oxford, UK. ³Department of Computational Biology and Medical Sciences, Graduate School of Frontier Sciences, The University of Tokyo, Tokyo, Japan. ⁴Analytic and Translational Genetics Unit, Massachusetts General Hospital, Boston, MA 02114, USA. ⁵Program in Medical and Population Genetics, Broad Institute of MIT and Harvard, Cambridge, MA 02142, USA. ⁶Department of Statistical Genetics, Osaka University Graduate School of Medicine, Suita 565-0871, Japan. ⁷Department of Genome Informatics, Graduate School of Medicine, University of Tokyo, Tokyo 113-0033, Japan. ⁸Laboratory for Systems Genetics, RIKEN Center for Integrative Medical Sciences, Kanagawa 230- 0045, Japan. ⁹Laboratory of Statistical Immunology, Immunology Frontier Research Center (WPI-IFReC), Osaka University, Suita 565-0871, Japan. ¹⁰National Center for Cardiovascular Diseases, Fuwai Hospital, Chinese Academy of Medical Sciences, 100037 Beijing, China. ¹¹Zhejiang CDC, Zhejiang, China. ¹²Department of Epidemiology & Biostatistics, School of Public Health, Peking University, Xueyuan Road, Haidian District, 100191 Beijing, China. ¹³Peking University Center for Public Health and Epidemic Preparedness and Response, 100191 Beijing, China. ¹⁴Key Laboratory of Epidemiology of Major Diseases (Peking University), Ministry of Education, 100191 Beijing, China. ¹⁵China National Center For Food Safety Risk Assessment, Beijing, China. ¹⁶These authors contributed equally: Alfred Pozarickij, Wei Gan, Kuang Lin. ¹⁷These authors jointly supervised this work: Zhengming Chen, Liming Li, Iona Y. Millwood, Robin G. Walters. **A list of members and their affiliations appears in the Supplementary Information.
✉e-mail: lmleeph@vip.163.com; robin.walters@ndph.ox.ac.uk

## China Kadoorie Biobank Collaborative Group

Alfred Pozarickij[1,16], Kuang Lin[1,16], Robert Clarke[1], Derrick Bennett[1], Huaidong Du[1], Yiping Chen[1], Ling Yang[1], Daniel Avery[1], Yu Guo[10], Min Yu[11], Canqing Yu[12,13,14], Dan Schmidt Valle[1], Jun Lv[12,13,14], Junshi Chen[15], Richard Peto[1], Rory Collins[1], Liming Li[12,13,14,17] ✉, Zhengming Chen[1,17], Iona Y. Millwood[1,17] & Robin G. Walters[1,17] ✉

