## [Peer Review File · Nature Communications]

Causal relevance of different blood pressure traits on risk of cardiovascular diseases: GWAS and Mendelian randomisation in 100,000 Chinese adultsREVIEWER COMMENTS

Reviewer #1 (Remarks to the Author):

The working group performed GWAS of blood pressure (BP) traits and then examined the causal link with cardiovascular diseases (CVD) using Mendelian randomization (MR) methods in Chinese adults. The manuscript identified 128 non-overlapping regions and ~ 500 separate trait associations. They also found that the effect sizes and heritability were much higher in the Chinese dataset compared to BBJ and European populations. MR analysis indicated a strong causal link between BP and CVD risk. Overall, this is an interesting study, representing an important progress in elucidating the genetics of BP, particularly in the Chinese population that accounts for more than 15% of world population. I have the following comments.

1. I am concerned by the MR analysis. The authors used the polygenic risk score (PRS) with variants/effect-sizes from the current study to examine the causal effect size in the same data set. Would this cause bias given the fact that BP is a strong risk factor for CVD (worthwhile to be reported in this nice cohort study)? In addition, standard MR methods based on summary statistics have a step to examine horizontal pleiotropy, a major concern for MR analysis. The PRS based method without examining this effect may be subject to bias due to this. Furthermore, I would also suggest performing MR analysis using summary statistics from independent datasets (e.g., BBJ, although with very different estimates of heritability). MVMR methods are also available for multiple instruments.
2. Reported higher heritability in CKB than in BBJ and IBP. Some additional analyses would be very helpful to better understand the difference. For example, the anti-BP treatment is much lower in CKB than in the other populations (Supplementary table). The authors are encouraged to do stratified analysis to see whether the treatment modifies the genetic effect to BP. Further, stratified analysis by age group (or other variables) may also be useful. SNP-environment interactions shall be examined for these genome-wide significant variants.
3. Popcorn analysis between East Asian and EUR: is the reported genetic correlation based on effect sizes (beta) or based on phenotypic variances of individual SNP? If consistent, they need to say so.
4. Summary statistics of 4 BP traits are strongly encouraged to be posted online to benefit the genetic research community.
5. Author might consider meta-analyzing CKB and BBJ to reveal further loci. But this is optional, particularly if the summary statistics for CKB is made public.
6. Figure 1b, s.e. for the estimated genetic correlation shall be added.
7. "17 out of 23 available lead variants were directionally consistent in ICBP, with 9 replicated at 5% FDR" page 7, is not clear.

Reviewer #2 (Remarks to the Author):

This work applied GWAS to four blood pressure traits (SBP, DBP, MAP and PP) in a data of Chinese adults. As most blood pressure (BP) GWAS to date has been limited to SBP and DBP, the authors argue that analyses of MAP and PP provides new insights to high BP etiology. Furthermore, they perform causal analyses based on mendelian randomization to assess the causal role of BP traits on cardiovascular and neurovascular diseases. Although the work provides an extensive treatment on the area of BP genetics, there is still a large room for improvement in their causal analyses. Especially, the lack of formal causal language (based on either structural causal model or potential outcome) makes it difficult for any thorough evaluation of the work.

Major comments:

1. From the title and the content, one can read that Mendelian randomization (MR), a causal analysis, is the main analysis of the study. Nevertheless, the study lacks essential components of modern causal analyses in which Judea Pearl's structural causal model (SCM) or Neyman-Rubin potential outcome (PO) should be explicitly stated and used in the analysis. Among the two, presenting the directed acyclic graph (DAG) within the SCM framework should be more familiar to the readers of epidemiology and genetics. The causal relationships between variables should be depicted in DAG and should be presented as a main figure that guides the analyses throughout the article. If not, any causal inference remains opaque to the reader.

First, one should provide the rationale of adjusting for body mass index (BMI) within the SCM framework. For example, the authors have reported that adjusting BMI significantly reduces the signal at the FTO region. Depending on the DAG, BMI can be either a confounder of or a mediator. In the former case, the controlled effect should be interpreted as the direct effect while in the latter case, it should be interpreted as the direct effect.

Second, as the authors perform MR based on BMI-adjusted GWAS, the interpretation of the estimated causal effect should also depend on the causal relationship of BMI and other variables in the DAG.

2. MAP and PP are essentially not truly new traits to be analyzed. MAP is calculated as a weighted mean of SBP and DBP (where the weights are 1/3 and 2/3 respectively) and PP is the difference between the two. Therefore, a significant genetic correlation (as well as phenotypic correlation) is totally expected. Moreover, it should be noted that assuming the linear model, the effect size and the P-values of MAP and PP GWAS can be obtained using the SBP and DBP GWAS. For example, the PP effect size should be the difference between the SBP and the DBP effect size which is a provable consequence of subtracting the regression equation of the former from the regression equation of the latter BP trait.

The collinearities arising from the definitions of the trait also raise a challenge in constructing a causal DAG. Is it the DBP and SBP that causes PP and MAP or the other way around? Answering these questions raises concerns on the interpretation of the subsequent MR analyses. To make things transparent, a DAG is inevitable, and the authors should provide all possible (but reasonable) DAGs and the following interpretations of their analyses depending on the DAG.

3. If the authors want to demonstrate the relatively large effect size and heritability found in their cohort as a main result, they should provide some putative explanations for their observations. For example, they have listed several non-genetic variables that alters BP in the beginning of the manuscript. Therefore, one possible hypothesis that explains the larger heritability would be that those non-genetic variables are less variable in their cohort compared to others.

Reviewer #3 (Remarks to the Author):

The authors report GWAS studies on blood pressure traits using a Chinese population biobank. The sample size used is the biggest from a Chinese population I could find in the literature.

The authors employed standard programs and methods in GWAS research, and 40 loci are reported which have not been previously reported for any ethnicity, reinforcing the view that different ethnicities need to be studied for a full understanding of genetics of disease.

I would like to see some points made clearer or expanded on:

1) There are other studies of BP in Chinese populations (e.g. PMID 25249183). Standard replication in a population of the same ethnicity would help to know if the novel findings are real and possibly population-specific. Would it be possible to perform replication with a Chinese dataset?

2) The heritability in BBJ is much lower than the two other populations considered, but the analyses run in BBJ take into consideration a different set of covariates. One that intrigues me is "status of 47 target diseases". Do the authors mean there were 47 covariates included, each to indicate if individuals were cases or controls of 47 diseases? Why was such adjustment not performed in the CKB analysis? Could this dilute the variance explained, by lowering the effect estimates of the variants?

- 3) Suppl. Table 2: the sample size matches UKB data, not ICBP, which has a sample size of ~299k . Did you then use ICBP data, UKB data or merged (summary stats are available for the joint analysis ICBP+UKB [N~757k], therefore the doubt)?
- 4) I miss some explanation about the novel variants. There's no description of loci, no table with the novel variants, no column in the supplementary tables to indicate which ones are novel, in order to double-check they are indeed novel.
- 5) Following up on the previous point, I saw no (e/p/s)QTL analyses, which would be interesting to hypothesize on mechanisms of these novel variants.

Minor

- 6) Page 9: 1 SD can't be 156 mmHg, it seems to be mean + 1SD, but it's not what the text says.
- 7) Typo: "in this instances" (page 22)

“Causal relevance of different blood pressure traits on risk of cardiovascular diseases: GWAS and Mendelian randomisation in 100,000 Chinese adults” (NCOMMS-23-03932-T)

Response to reviewers:

We have revised the manuscript in light of helpful comments from the three reviewers. The reviewers' verbatim comments are shown below in bold followed by our responses. Page and line numbers indicating where changes have been made refer to the new clean version of the manuscript without tracked changes.

We have one general response relevant to all reviewers: in the MR sections of this work, we were not primarily attempting to demonstrate a causal relationship between blood pressure and CVD – this has been comprehensively addressed by our group in a separate recent publication¹. Instead, our primary objective in the MR sections of this work was to discriminate which of the different strongly-correlated aspects of blood pressure might be responsible for this established causality, by comparing the associations of 4 different BP traits and analysing them using MVMR. We regret that this objective was not sufficiently clear in the previous version. We have updated the text throughout, and included a new explanatory Figure, to more clearly identify this key objective of our MR analyses.

Reviewer #1

1. I am concerned by the MR analysis. The authors used the polygenic risk score (PRS) with variants/effect-sizes from the current study to examine the causal effect size in the same data set. Would this cause bias given the fact that BP is a strong risk factor for CVD (worthwhile to be reported in this nice cohort study)?

We agree that using internal weights to construct genetic scores can potentially cause bias due to overfitting (leading to bias towards the observational association). While we believe this is unlikely to impact our comparison of different highly-correlated BP traits (biases would largely cancel out), we agree that it is better to avoid such over-fitting. Therefore, we have constructed revised genetic scores by performing 100-fold jack-knifing, so that the score for any given individual is not derived using data from that individual. We describe this change on page 9 lines 213-214 and page 26 lines 587-591. We observed a small reduction in the variance explained for each BP trait by the resulting updated scores (e.g. 3.2% vs 3.1% for SBP and 3.3% vs 3.1% DBP). We present the updated findings in Supplementary Table 19 and have modified the text on page 9 line 215 accordingly.

The new jack-knifed scores were then used in updated MR and MVMR analyses. Both sets of results were near-identical to the original findings (e.g. association of pulse pressure with carotid plaque is now OR 3.40 [2.56-4.51] vs 3.38 [2.57-4.46] previously). We present the updated findings in Figures 3 and 4 and Supplementary Figures 15-17. We have modified the text on pages 9-11 to take into account these changes.

In addition, standard MR methods based on summary statistics have a step to examine horizontal pleiotropy, a major concern for MR analysis. The PRS based method without examining this effect may be subject to bias due to this.

We agree that this should be included. Using either pre-existing CKB GWAS for the disease endpoints, or newly-derived effect sizes for carotid plaque, we performed MR-Egger (and the related IVW MR). We found no evidence for unbalanced pleiotropy, and the estimates using this method were consistent with those from our main analysis. We describe this on page 10 lines 231-235 with the results provided in Supplementary Table 20, and have updated the methods on page 27 lines 615-621.

Furthermore, I would also suggest performing MR analysis using summary statistics from independent datasets (e.g., BBJ, although with very different estimates of heritability).

As suggested by the reviewer, we obtained BBJ summary statistics for ischaemic stroke, intracerebral haemorrhage, and myocardial infarction (<https://pheweb.jp/>) and performed two-sample MR, as described in the online methods (page 27 lines 615-621). Effect sizes from the BBJ analysis were consistent with our CKB results, again with no evidence for unbalanced pleiotropy in MR-Egger analyses. We provide these two-sample MR results in Supplementary Table 20 and have added text describing these on page 10 lines 231-235.

MVMR methods are also available for multiple instruments.

We also attempted two-sample MVMR based on summary statistics from both CKB and BBJ summary statistics. In these analyses (which have less statistical power than our main analyses using individual level data), the instruments for several trait pairs had low conditional F-statistics and were subject to weak instrument bias. However, in those cases where instruments had $F\text{-cond} > 10$, the results were mostly consistent with those in the CKB analyses. However, for ICH risk these analyses have led us to make some minor changes to our conclusions. We provide these results in Supplementary Figure 18 and Supplementary Table 22 and compare them with our previous results on page 12 lines 281-290.

We were unable to attempt external replication of our finding of a causal association between pulse pressure and carotid plaque (CP), because summary statistics for CP are not available in BBJ (or other East Asian datasets). However, we performed a two-sample MVMR for CP within CKB. To avoid bias due to sample overlap, we estimated variant effect sizes for CP and re-estimated BP effect sizes in a sample that excluded individuals used to derive SNP effects for CP. These changes are described on page 12 lines 283-284 and page 27 lines 616-619, Supplementary Figure 18d and Supplementary Table 22.

2. Reported higher heritability in CKB than in BBJ and ICBP. Some additional analyses would be very helpful to better understand the difference. For example, the anti-BP treatment is much lower in CKB than in the other populations (Supplementary table). The authors are encouraged to do stratified analysis to see whether the treatment modifies the genetic effect to BP. Further, stratified analysis by age group (or other variables) may also be useful.

We agree that this difference is intriguing (having briefly discussed possible reasons in the previous version of the manuscript), and thank the reviewer for the suggested additional analyses. In common with other studies, including ICBP and BBJ, our GWAS attempted to account for the effects of anti-BP treatment by adjusting SBP/DBP by adding 15/10 mmHg respectively (see page 22 lines 496-498 in online methods). As in other studies, we had expected this would mostly account for the effect of medication.

Having reported in the previous version of the manuscript that heritability does not substantially differ across CKB study regions (Supplementary Figure 12 and Supplementary Table 15), we have as suggested extended this to other subgroups to assess possible differences in heritability by treatment, sex, and age (stratified into the groups described in our previous publication¹), which are now presented in Supplementary Table 16. Overall, while these stratified heritability estimates do not differ appreciably by sex or age, we find a substantial difference in heritability of SBP according to medication use (and a lesser difference in MAP). We describe these results (page 8, lines 193-201) and methods (page 24, lines 550-551), and have included discussion of them in relevant sections (page 16, lines 369-372).

SNP-environment interactions shall be examined for these genome-wide significant variants.

We thank the reviewer for this suggestion. We had already considered differences between urban and rural regions (see Supplementary Figure 12), and have extended this to examine effect modification by anti-hypertensive treatment status (shown in Supplementary Figure 13). We find that anti-BP medication use reduces variant effect size by 43% for each of SBP, DBP, MAP (but not PP). These results are described on page 8 lines 201-202 and discussed on page 16 lines 363-367.

We have discussed (page 16 lines 373-376) how, taken together, these effects of medication use on heritability and effect size provide a potential explanation for the observed population differences. This also highlights the benefits of performing analyses in largely medication-naïve populations such as CKB, where higher heritability and genetic effect sizes may be revealed.

3. Popcorn analysis between East Asian and EUR: is the reported genetic correlation based on effect sizes (beta) or based on phenotypic variances of individual SNP? If consistent, they need to say so.

Popcorn is based on the correlation of variant effect sizes from GWAS summary statistics. This has been clarified in the text (page 9 line 205).

4. Summary statistics of 4 BP traits are strongly encouraged to be posted online to benefit the genetic research community.

We are in the process of submitting summary statistics for the 4 BP traits (with and without BMI adjustment) to GWAS Catalog, and these will be released upon publication. We will also provide them on our CKB PheWeb browser at pheweb.ckbiobank.org.

5. Author might consider meta-analyzing CKB and BBJ to reveal further loci. But this is optional, particularly if the summary statistics for CKB is made public.

We agree that a future meta-analysis across East Asian cohorts would be of value (which could also include the recent results from Korean and Taiwanese biobanks). However, the observed very different effect sizes between populations poses a challenge for meta-analysis – for this reason we did not meta-analyse even those variants that were replicated in BBJ. We have had preliminary discussions with other East Asian cohorts about a future meta-analysis, but feel that this is out of the scope of the current paper.

6. Figure 1b, s.e. for the estimated genetic correlation shall be added.

We apologise for this oversight, we have now added the s.e. in Figure 1b and Supplementary Figure 7b.

7. "17 out of 23 available lead variants were directionally consistent in ICBP, with 9 replicated at 5% FDR" page 7, is not clear.

We apologise for not being clearer. We have changed the sentence to "out of 18 CKB lead variants for which ICBP summary statistics were available, 10 associations were directionally concordant and 4 reached significance at 5% FDR" (page 7 lines 173-176). Note that the total number of novel associations has been updated (reduced) in light of recent additional publications, leading to additional changes to Table 1, Supplementary Tables ST4, ST14, and Supplementary Figure 8.

Reviewer #2

1. From the title and the content, one can read that Mendelian randomization (MR), a causal analysis, is the main analysis of the study. Nevertheless, the study lacks essential components of modern causal analyses in which Judea Pearl's structural causal model (SCM) or Neyman-Rubin potential outcome (PO) should be explicitly stated and used in the analysis. Among the two, presenting the directed acyclic graph (DAG) within the SCM framework should be more familiar to the readers of epidemiology and genetics. The causal relationships between variables should be depicted in DAG and should be presented as a main figure that guides the analyses throughout the article. If not, any causal inference remains opaque to the reader.

As noted above, apart from the broad description of the genetic architecture of BP traits, the primary aim of the paper was not a formal test of the causal relevance of BP to CVD, which has been examined elsewhere¹, but to assess what aspects of BP are most important causally for disease outcomes. For this, we are applying the well-established MR framework, an approach very commonly used to assess causal relationships between genetically predicted exposure and the outcome², which will be very familiar to readers. To further clarify our causal framework and methodology which involves multiple BP traits, we have included an explanatory diagram for the MVMR analyses, which readers may be less familiar with. We include this as a Supplementary Figure 14 and have amended the discussion relating to this (page 10 lines 237-240).

First, one should provide the rationale of adjusting for body mass index (BMI) within the SCM framework. For example, the authors have reported that adjusting BMI significantly reduces the signal at the FTO region. Depending on the DAG, BMI can be either a confounder of or a mediator. In the former case, the controlled effect should be interpreted as the direct effect while in the latter case, it should be interpreted as the direct effect.

As stated in the manuscript in both results and discussion (page 5 lines 122-124, page 9 lines 208-209, page 13 lines 305-307), our primary reason for considering BMI was to enable direct comparability with previous studies (i.e. ICBP performed analyses adjusting for BMI, whereas no such adjustment was made by BBJ). While we very much agree that consideration of the interaction of BMI and BP is an important topic, this would require a wide range of additional analyses that are well beyond the scope of this paper.

Second, as the authors perform MR based on BMI-adjusted GWAS, the interpretation of the estimated causal effect should also depend on the causal relationship of BMI and other variables in the DAG.

While BMI influences BP, the key message from our analysis is that (with the exception of FTO, the BMI locus with by far the largest effect size), BMI has negligible impact on BP effect size for any of the BP trait-associated variants included in the MR analysis. Thus, apart from FTO, our BP-associated loci are not confounded by BMI and do not depend on BMI. Our primary analyses use results from BMI-adjusted GWAS to restrict any residual contribution to BP that might be due to effects of BMI.

2. MAP and PP are essentially not truly new traits to be analyzed. MAP is calculated as a weighted mean of SBP and DBP (where the weights are 1/3 and 2/3 respectively) and PP is the difference between the two. Therefore, a significant genetic correlation (as well as phenotypic correlation) is totally expected

The observational and genetic correlation between the different BP traits is central to the question being addressed in this study. Disease risk is very commonly reported in relation to SBP (and perhaps DBP), but it is far from established that these are the key aspects of BP for disease risk. While MAP and PP are derived as linear combinations of measured SBP and DBP, they reflect different aspects of physiology and vascular anatomy (as discussed in the manuscript at page 3 lines 78-83 and page 14 lines 328-335). It is of great etiological and medical interest to identify which of these traits is most strongly associated with disease risk, and which trait has the “true” causal effect.

Moreover, it should be noted that assuming the linear model, the effect size and the P-values of MAP and PP GWAS can be obtained using the SBP and DBP GWAS. For example, the PP effect size should be the difference between the SBP and the DBP effect size which is a provable consequence of subtracting the regression equation of the former from the regression equation or the latter BP trait.

Statistically, MAP effect sizes are not a fixed linear combination of SBP and DBP effect sizes, nor are PP effect sizes the difference between SBP and DBP effect sizes. In order to understand the role of BP in disease risk, it is necessary to analyse these traits separately. Indeed the identification of loci specifically associated with PP and MAP, and not SBP and DBP, reinforces that one cannot reliably estimate the effect size and P-values of MAP and PP GWAS using SBP and DBP GWAS.

The collinearities arising from the definitions of the trait also raise a challenge in constructing a causal DAG. Is it the DBP and SBP that causes PP and MAP or the other way around? Answering these questions raises concerns on the interpretation of the subsequent MR analyses. To make things transparent, a DAG is inevitable, and the authors should provide all possible (but reasonable) DAGs and the following interpretations of their analyses depending on the DAG.

Each of these different BP measures reflect different physiological facets of the overall pattern of BP that occurs during the cardiac cycle, and as such it is appropriate to consider them as distinct measures, rather than causing or being caused by each other. While MAP and PP are not directly recorded but are traits derived from recorded SBP and DBP, these have nevertheless been used widely, in addition to SBP/DBP, in clinical and epidemiological studies. Understanding their genetic determinants may shed new insights into genetic architecture of blood pressure. As discussed in the manuscript (page 4 lines 93-95), the genetic determinants for PP and MAP are still less well understood than those for SBP and DBP. As an example, the largest study of blood pressure did not include MAP³. Indeed, our GWAS showed that by including these traits we can identify additional genetic determinants not associated with SBP and DBP (Supplementary Table 4).

In this manuscript, we have performed MVMR precisely to disentangle the causal effects of these blood pressure phenotypes on several CVD outcomes. As noted above, we acknowledge that our MVMR analyses of the complicated relationship between traits would benefit from an illustration (Supplementary Figure 14).

3. If the authors want to demonstrate the relatively large effect size and heritability found in their cohort as a main result, they should provide some putative explanations for their observations. For example, they have listed several non-genetic variables that alters BP in the beginning of the manuscript. Therefore, one possible hypothesis that explains the larger heritability would be that those non-genetic variables are less variable in their cohort compared to others.

We agree that non-genetic factors may play an important role in the different heritability estimates found in different cohorts, and we had included a detailed discussion addressing this point, highlighting differences in lifestyle (including salt intake), the population-based nature of CKB compared to hospital-based BBJ, and extent of BP medication use. We have updated and expanded this discussion, including the analyses prompted by Reviewer 1 which show a substantial impact of medication on SBP heritability (Supplementary Table 16). We do agree, however, that lower environmental variability is a potential source of increase d heritability, and have included that point in the discussion (page 16 lines 367-368).

Reviewer #3

1. There are other studies of BP in Chinese populations (e.g. PMID 25249183). Standard replication in a population of the same ethnicity would help to know if the novel findings are real and possibly population-specific. Would it be possible to perform replication with a Chinese dataset?

We agree that replication in a closely matched cohort would be of value, and this motivated us to seek replication in a Japanese ancestry biobank (Supplementary Table 14). Unfortunately, the specific suggested study (PMID 25249183) only considered SBP and DBP, and summary statistics are not available; furthermore, it had a comparatively small sample size (discovery N=11,816) so would have very limited power for replication. We have added a comment about the value of more closely matched replication to the study limitation section (page 20 lines 452-454).

2. The heritability in BBJ is much lower than the two other populations considered, but the analyses run in BBJ take into consideration a different set of covariates. One that intrigues me is "status of 47 target diseases". Do the authors mean there were 47 covariates included, each to indicate if individuals were cases or controls of 47 diseases? Why was such adjustment not performed in the CKB analysis? Could this dilute the variance explained, by lowering the effect estimates of the variants?

BBJ is a hospital-based study, involving primarily patients admitted to hospital with 47 different types of conditions. To avoid potential biases, they therefore routinely adjust for disease ascertainment in their analyses, as documented in many BBJ publications⁴. By comparison, this approach is not necessary in CKB. CKB is a population-based cohort study, involving relatively healthy individuals from the general community. As described in the methods (page 22 lines 501-503), we performed all adjustments for covariates and data transformations in the full CKB cohort (N ~ 500K), prior to genetic association analyses, to

avoid any potential ascertainment bias.⁵ Moreover, our heritability estimates are similar to those more recently reported by Taiwan Biobank⁶, which is also a population-based cohort study. We have sought to clarify these details on page 15 lines 344-345 and pages 16-17 lines 373-378.

3. Suppl. Table 2: the sample size matches UKB data, not ICBP, which has a sample size of ~299k. Did you then use ICBP data, UKB data or merged (summary stats are available for the joint analysis ICBP+UKB [N~757k], therefore the doubt)?

Thank you for noting this error. As noted by the reviewer the sample size should be 757,601 as the summary statistics involved the joint analysis of both ICBP and UKB. We have corrected Supplementary Table 2 accordingly.

4. I miss some explanation about the novel variants. There's no description of loci, no table with the novel variants, no column in the supplementary tables to indicate which ones are novel, in order to double-check they are indeed novel.

We apologise for not providing this information sufficiently clearly. For clarification, the list of novel variants was presented in Supplementary Table 14. We have now updated identification of novel variants in line with additional data submitted to GWAS Catalog. The number of novel variants have decreased to 74 and we have modified Supplementary Tables 4 and 14, and Supplementary Figures 2,3,4,5,8,10, as well as text in the manuscript (page 2 line 61, page 6 line 136, page 7 line 171, page 13 line 295, page 25 line 567) to take this change into account. We have in addition modified the text (page 6 line 138) to make it clearer where the novel variants can be located.

5. Following up on the previous point, I saw no (e/p/s)QTL analyses, which would be interesting to hypothesize on mechanisms of these novel variants.

We are grateful for this suggestion. We agree that it would be of value to explore the mechanisms underlying these novel associations by integrating results with other omics resources. However, at present we are limited by the lack of suitable EAS datasets for these various analyses, e.g. GTEx only has 12 donors of Asian descent. We are currently working on integrating pQTL data with the 4 BP traits considered in this manuscript. However, we believe analyses such as these are outside the scope of the current paper, and would not contribute to the intended main findings of the paper. Furthermore, the novel loci are skewed towards under-studied MAP, and are potentially unrepresentative. We mention this suggestion in the study limitation and future implications sections in the updated manuscript (page 20 lines 454-457).

6. Page 9: 1 SD can't be 156 mmHg, it seems to be mean + 1SD, but it's not what the text says.

We thank the reviewer for spotting this error. This was supposed to be the value for 1SD (22.3 mmHg), and we have corrected the error (page 9 line 220).

7. Typo: "in this instances" (page 22)

We have corrected this typo (now page 24 line 558), thank you.

References

1. Clarke, R. *et al.* Genetically Predicted Differences in Systolic Blood Pressure and Risk of Cardiovascular and Noncardiovascular Diseases: A Mendelian Randomization Study in Chinese Adults. *Hypertension* **80**, 566-576 (2023).
2. Sanderson, E. *et al.* Mendelian randomization. *Nature Reviews Methods Primers* **2**, 6 (2022).
3. Evangelou, E. *et al.* Genetic analysis of over 1 million people identifies 535 new loci associated with blood pressure traits. *Nature Genetics* **50**, 1412-1425 (2018).
4. Nagai, A. *et al.* Overview of the BioBank Japan Project: Study design and profile. *J Epidemiol* **27**, S2-s8 (2017).
5. Walters, R.G. *et al.* Genotyping and population characteristics of the China Kadoorie Biobank. *Cell Genom* **3**, 100361 (2023).
6. Chen, C.-Y. *et al.* Analysis across Taiwan Biobank, Biobank Japan and UK Biobank identifies hundreds of novel loci for 36 quantitative traits. *medRxiv*, 2021.04.12.21255236 (2021).

REVIEWER COMMENTS

Reviewer #1 (Remarks to the Author):

My comments have been addressed successfully.

Reviewer #2 (Remarks to the Author):

The comments are attached as a file.

Reviewer #3 (Remarks to the Author):

The authors answered my questions satisfactorily.

As the authors disagree with us in many ways, I think it's better to narrow down the points. Firstly, I still doubt that the GWAS on two traits (PP and MAP) are truly novel. Secondly, drawing a causal graph is not a big deal that force a substantial amount of reanalysis. I don't get the authors claim that it's out of the scope.

1. Linear regression of linearly transformed traits are indeed exchangeable. I'm providing a simple proof for this claim.

Let \mathbf{y}_1 and \mathbf{y}_2 be the two traits of interest. The bold face indicates that they are $n \times 1$ -vectors of n -samples. Let \mathbf{X} be the $n \times p$ matrix of explanatory variables. Given that $\mathbf{X}^t\mathbf{X}$ is full-rank, the regression coefficient is $\hat{\beta}_1 = (\mathbf{X}^t\mathbf{X})^{-1}\mathbf{X}^t\mathbf{y}_1$ and $\hat{\beta}_2 = (\mathbf{X}^t\mathbf{X})^{-1}\mathbf{X}^t\mathbf{y}_2$. Therefore, for a linear transformed trait $a\mathbf{y}_1 + b\mathbf{y}_2$ (a and b are constants) of trait 1 and trait 2, the regression coefficient is

$$(\mathbf{X}^t\mathbf{X})^{-1}\mathbf{X}^t(a\mathbf{y}_1 + b\mathbf{y}_2) = a(\mathbf{X}^t\mathbf{X})^{-1}\mathbf{X}^t\mathbf{y}_1 + b(\mathbf{X}^t\mathbf{X})^{-1}\mathbf{X}^t\mathbf{y}_2 = a\hat{\beta}_1 + b\hat{\beta}_2$$

It shows that the summary statistics of PP and MAP can be deduced from SBP and DBP summary statistics.

In practice, traits are z-transformed to have equal variance which might change the constants a and b . Nevertheless, the difference is only up to a constant so quantities like P -values are conserved. Analytic derivations of standard errors and P -values also follow from basic algebraic formula of linear regression.

2. There aren't much difference between genetic correlation and phenotypic correlation. It follows the same mathematical arithmetics. Let a, b, c and d be constants as before. We have two linearly transformed traits $z_1 = a\mathbf{y}_1 + c\mathbf{y}_2$ and $z_2 = c\mathbf{y}_1 + d\mathbf{y}_2$. Then following the previous argument, for any locus k , the coefficients are $a\beta_{1k} + b\beta_{2k}$ and $c\beta_{1k} + d\beta_{2k}$.

Then the genetic covariance (=the genetic correlation prior to division by heritability), which is defined as the covariance of coefficients under the random effects assumption are

$$Cov(a\beta_{1k} + b\beta_{2k}, c\beta_{1k} + d\beta_{2k}) = acVar(\beta_{1k}) + bdVar(\beta_{2k}) + (ad + bc)Cov(\beta_{1k}, \beta_{2k})$$

The equaiton shows that the genetic covariance of the two linearly transformed traits can be deduced from the heritability and genetic correlation of two original traits. Hence, any non-zero genetic correlation between the suggested traits are totally expected. Do we learn anything new beyond this expected level of genetic correlation?

3. The schematics well describe the most basic setting where MVMR can be applied. As you see in the original MVMR paper (PMC4325677), there are two DAGs in Figure 3 where the first figure (Fig 3a) is identical to Supp Figure 14 of the manuscript and the second figure being a generalization (Fig 3b). Here's the figure from the original

paper.

Figure 3. Causal directed acyclic graph illustrating multivariable Mendelian randomization in associations between variants G_1 , G_2 , and G_3 , risk factors X_1 and X_2 , and outcome Y . Confounders U_1 and U_2 are assumed to be unknown. A) Risk factors are causally independent (no causal effects between X_1 and X_2); B) risk factors are causally dependent (X_1 has a causal effect on X_2).

In the case of four distinct BP traits, it falls into the second DAG because the rank of the four traits is actually two in a linear algebraic sense (two traits are deterministic function of the other two). Therefore, Supp Fig 14 is an incomplete and a potentially misleading diagram without the arrows between the risk factors.

Indeed, the original authors of MVMR includes an analysis depicting the impact of causally dependent risk factors. They show that MVMR only identifies the “direct causal effect” and not the “total causal effect”. I’m copy-pasting the paragraph of the original paper:

Table 2 shows the mean estimates of the causal parameters derived from each of the methods. Aside from the regression-based method, which produces widely varying results, we see that the estimates do not change substantially as the parameters vary. This indicates that the 2SLS and likelihood-based methods estimate the direct causal effect of each risk factor on the outcome, not including paths operating via the other risk factors. This can lead to misleading conclusions about the total effects of the variables. For example, when $\alpha_{X2} = 0$ and $\alpha_{X3} = 0.5$, the total causal effect of X_3 on Y is $\beta_3 + \alpha_{X1}\beta_1 + \frac{1}{4} 0:1 + 0:5 \times 0:3 + \frac{1}{4} 0:05$ (including the path operating via X_1). The mean estimates from the 2SLS and likelihood-based methods are in the opposite direction of the true total effects.

As I pointed out in the earlier round of the review, the authors should be explicit on which BP trait is the cause of the other. It might even be more plausible to consider two *latent* BP traits that causes the four traits (DBP, SBP, MAP and PP). As I’m not a domain expert in cardiology, I believe that the authors have more expertise to justify either of these causal dependence between the four BP traits.

“Causal relevance of different blood pressure traits on risk of cardiovascular diseases: GWAS and Mendelian randomisation in 100,000 Chinese adults” (NCOMMS-23-03932A)

We are pleased that Reviewer #1 and Reviewer #3 have no further comments on our manuscript. The further comments by Reviewer #2 are in large part, we would strongly argue, based on a misunderstanding of the nature of blood pressure (BP) traits – as the reviewer acknowledges, they are not a “domain expert in cardiology”. The key point, which we have attempted to make clear in the manuscript in several places, is that different measures of BP are not intrinsic physiological phenotypes, but are the outputs of particular ways of measuring and recording complex fluctuations in BP during the cardiac cycle, which are in turn influenced by more fundamental aspects of vascular and/or cardiac physiology. Thus, SBP and DBP do not “cause” PP and MAP, they are simply the most readily measured parameters. For example: unlike SBP and DBP, PP is a major indicator of vascular aging and arterial stiffness¹; MAP is an indicator of blood flow and is influenced by arteriolar constriction and dilation, for which SBP and DBP individually are less useful measures²; and the results from our MVMR analyses indicate that only PP is associated with carotid plaque, clearly demonstrating that we get different information by using different BP traits. We believe this point is already sufficiently discussed (e.g. at p3 lines 78-83 and p14 lines 330-337).

The remaining comments from Reviewer #2 are mostly about the concept of the paper, but we have sought to add text to address any specific comments, as follows. The reviewer’s verbatim comments are shown below in bold followed by our responses in italics.

As the authors disagree with us in many ways, I think it’s better to narrow down the points. Firstly, I still doubt that the GWAS on two traits (PP and MAP) are truly novel.

We agree that the GWAS for MAP and PP are not novel per se. Indeed, they have been widely investigated, including in numerous previous genomics studies. As noted above, they reflect different aspects of physiology from those reflected by SBP and DBP, and thus merit consideration when attempting to determine the way(s) in which BP influences CVD risk. The benefit of considering all four BP traits is apparent from, for instance, the MVMR results in which using multiple trait-specific GWAS signals enables identification of different patterns of results for different disease outcomes. These points are already addressed in the manuscript at p12 lines 276-282.

Secondly, drawing a causal graph is not a big deal that force a substantial amount of reanalysis. I don’t get the authors claim that it’s out of the scope.

Our previous comment about a formal causal analysis being beyond the scope of this paper was in response to the suggestion of considering the effects of BMI. A DAG that included this and other traits that potentially mediate the effects of variants on BP (including e.g. arterial stiffness) and the associated causal analysis would be highly complex and would not help to address our analysis in this manuscript of the relationships between BP traits and CVD.

Linear regression of linearly transformed traits are indeed exchangeable. I’m providing a simple proof for this claim. The proof shows that the summary statistics of PP and MAP can be deduced from SBP and DBP summary statistics. In practice, traits are z-transformed to have equal variance which might change the constants (a and b). Nevertheless, the difference is only up to a constant so quantities like P -values are conserved. Analytic derivations of standard errors and P -values also follow from basic algebraic formula of linear regression.

We agree that the effect size estimates for the different BP traits reflect the algebraic relationships between them (as can be demonstrated directly by comparison of variant effect sizes from the corresponding GWAS). However, it does not follow that P-values follow an equivalent relationship – for instance, the separate summary statistics for SBP and DBP do not account for the covariance between these traits, so that GWAS of PP has the potential to identify significant associations that cannot be derived from the SBP and DBP GWAS results. Thus, conducting GWAS for all 4 traits leads to discovery of a much larger number of associated variants, with many variants only identified for PP and/or MAP (see Figure 1). These points are already addressed in the manuscript at p15 lines 340-342, but we have now added a short note about the arithmetical relationship between traits and noted that a large proportion of loci are associated with only one trait despite this (p6 lines 151-153).

There aren't much difference between genetic correlation and phenotypic correlation. It follows the same mathematical arithmetics. The equation shows that the genetic covariance of the two linearly transformed traits can be deduced from the heritability and genetic correlation of two original traits. Hence, any non-zero genetic correlation between the suggested traits are totally expected. Do we learn anything new beyond this expected level of genetic correlation?

We agree that non-zero genetic correlation of phenotypically correlated traits is unsurprising given their arithmetic relationship, and have added a brief comment on this (p15 line 342). However, we do not make any strong claims about the genetic correlation of BP traits and we would have been criticised had we not included these estimates in the paper since we aim to understand the genetic architecture of BP in Chinese individuals and compare it with other populations. In addition, estimates of genetic correlations are required for the MVMR analyses and strong correlations underlie the low conditional F-statistics. Inclusion of the genetic correlation analyses is, therefore, relevant for discussion of the difficulties in discriminating between these highly correlated BP traits when it comes to performing MR and emphasises why it cannot be assumed that, for instance, an apparent genetic influence of SBP on disease actually reflects a true causal effect (see SBP effect in Figure 3 and Figure 4).

The schematics well describe the most basic setting where MVMR can be applied. As you see in the original MVMR paper (PMC4325677), there are two DAGs in Figure 3 where the first figure (Fig 3a) is identical to Supp Figure 14 of the manuscript and the second figure being a generalization (fig 3b). In the case of four distinct BP traits, it falls into the second DAG because the rank of the four traits is actually two in a linear algebraic sense (two traits are deterministic function of the other two). Therefore, Supp Fig 14 is an incomplete and potentially misleading diagram without the arrows between the risk factors. Indeed, the original authors of MVMR includes an analysis depicting the impact of causally dependent risk factors. They show that MVMR only identifies the “direct causal effect” and not the “total causal effect”. As I pointed out in the earlier round of the review, the authors should be explicit about which BP trait is the cause of the other.

As noted above, we disagree that we “should be explicit about which BP trait is the cause of the other”. Rather, we argue that none of them being directly “caused” by any of the other traits (see p14 lines 330-337), since all of them are manifestations of different biological and physiological aspects of blood pressure – i.e. we do not believe this concept applies to our analyses. However, we have no objection to modifying Supplementary Figure 14 to reflect the more general case where trait 1 potentially directly influences trait 2, and have added an arrow to denote this.

We are unclear what point the reviewer is intending with the quoted section of text. This comes from the section of the paper concerning 2-sample MVMR. For these analyses, we use the package written by the authors of the quoted paper, so we believe the issue highlighted does not apply.

It might even be more plausible to consider two *latent* BP traits that causes the four traits (DBP, SBP, MAP and PP). As I'm not a domain expert in cardiology, I believe that the authors have more expertise to justify either of these causal dependence between the four BP traits.

The concept of (many more than two) "latent" traits precisely corresponds to the idea that each BP trait is a reflection of the combined effects of multiple underlying biological and physiological processes. However, as noted above, consideration of such a complex causal model lies well outside the scope of this manuscript.

References

1. Said, M.A., Eppinga, R.N., Lipsic, E., Verweij, N. & Harst, P.v.d. Relationship of Arterial Stiffness Index and Pulse Pressure With Cardiovascular Disease and Mortality. *Journal of the American Heart Association* **7**, e007621 (2018).
2. <https://derangedphysiology.com/main/cicm-primary-exam/required-reading/cardiovascular-system/Chapter%20035/systolic-diastolic-and-mean-arterial-blood-pressure>

REVIEWERS' COMMENTS

Reviewer #2 (Remarks to the Author):

The authors addressed most of my concerns. I only have a minor comment. If the authors' argument is that measured BP traits reflect an underlying (but not directly measured) physiological character, a causal claim that the measured quantity changes the disease risk must be taken carefully. Vanderweele has a neat paper on this topic (https://journals.lww.com/epidem/fulltext/2022/01000/constructed_measures_and_causal_inference__towards.17.aspx). A few lines discussing the caveats in the conclusion/discussion section will suffice.